

# Long-term monitoring (1953-2019) of geomorphologically active sections on LIA lateral moraines under changing meteorological conditions

Moritz Altmann[1], Madlene Pfeiffer[2], Florian Haas[1], Jakob Rom[1], Fabian Fleischer[1], Tobias Heckmann[1], Livia Piermattei[1,3], Michael Wimmer[4], Lukas Braun[5], Manuel Stark[1], Sarah Betz-Nutz[1], Michael Becht[1]

[1]Department of Physical Geography, Catholic University of Eichstätt-Ingolstadt, Eichstätt, 85072, Germany
[2]Institute of Geography, University of Bremen, Bremen, 28359, Germany
[3]Swiss Federal Institute for Forest, Snow and Landscape Research (WSL), Birmensdorf, 8903, Switzerland
[4]Department of Geodesy and Geoinformation, TU Wien, Vienna, 1040, Austria
[5]Institute of Mathematics, Albert Ludwig University of Freiburg, Freiburg, 79104, Germany

*Correspondence to*: Moritz Altmann (MAltmann@ku.de)

**Abstract.** We show a long-term erosion monitoring of several geomorphologically active gully systems on Little Ice Age lateral moraines in the central Eastern Alps covering a total time period from 1953 to 2019 including several survey periods in order to identify corresponding morphodynamic trends. For the implementation, DEM of Differences were calculated based on multitemporal high-resolution digital elevation models from historical aerial images (generated by structure-from-motion photogrammetry with multi-view-stereo) and light detection and ranging from airborne platforms. Two approaches were implemented to achieve the corresponding objectives. First, by calculating linear regression models using the accumulated sediment yield and the corresponding catchment area (on a log-log scale), the range of the variability of the spatial distribution of erosion values within the areas of interest is shown. Secondly, we use volume calculations to determine the total/mean sediment output (and erosion rates) of the entire areas of interest. Subsequently, a comparison is made between the areas of interest and the epochs of both approaches. Based on the slopes of the calculated regression lines, it could be shown that the highest range of the variability of sediment yield within all areas of interest is in the first epoch (mainly 1950s to 1970s), as in some areas of interest sediment yield per square metre increases clearly more (regression lines with slopes up to 1.5), which in the later epochs (1970s to mid-2000s and mid-2000s to 2017/2019) generally decreases in 10 out of 12 cases (regression lines with slopes around 1). However, even in the areas of interest with an increase in the variability of sediment yield over time, the earlier high variabilities are no longer reached. This means that the spatial pattern of erosion in the gully heads changes over time as it becomes more uniform. Furthermore, using sediment volume calculations and corresponding erosion rates, we show a generally decreasing trend in geomorphic activity (amount of sediment yield) between the different epochs in 10 out of 12 areas of interest, while 2 areas of interest show an opposite trend where morphodynamics increase and remain at the same level. Finally, we summarise the results of long-term changes in the morphodynamics of geomorphologically active areas on lateral moraines by presenting the "sediment activity concept", which, in contrast to theoretical models, is based on actually calculated erosion. The level of geomorphic activity depends strongly on the characteristics of the areas of interest,





such as size, slope length and slope gradient, some of which are associated with deeply incised gullies. It is noticeable that
especially areas with decades of dead ice influence in the lower slope area show high geomorphic activity. Furthermore, we
show that system-internal factors as well as the general paraglacial adjustment process have a greater influence on long-term
morphodynamics than changing external weather and climate conditions, which, however, had a slight impact mainly in the
last, i.e. most recent epoch (mid-2000s to 2017/2019) and may have led to an increase in erosion at the areas of interest.
**Keywords:** Airborne Laser Scanning (ALS), DEM of Difference (DoD), historical aerial images (HAI), gully erosion, Little
Ice Age (LIA) lateral moraines, modelling, Structure-from-Motion (SfM) photogrammetry, proglacial areas, Weather Research
and Forecasting (WRF) model
**1 Introduction**
Since the end of the Little Ice Age (LIA) around 1850 and the strong global warming of the last decades (IPCC, 2021; Pepin
et al., 2022), proglacial areas play a special role in the current landscape changes of high alpine geosystems, as such areas are
strongly increasing due to the ongoing retreat of the glaciers (Deline et al., 2015; Heckmann and Morche, 2019; Haeberli and
Whiteman, 2021). The melting of glaciers leads to the release of unstable sediment sources, which are subsequently exposed
to several geomorphological slope processes, which can lead to high erosion rates.
The relationship between this glacier melt and slope instability has been subject of research for several decades. Church and
Ryder (1972) were the first to develop a theoretical model ("paraglacial concept") to describe future landscape change
throughout a proglacial area and defined the phase of transition as the paraglacial period, during which paraglacial processes
(non-glacial processes) occur. After a period of high geomorphic activity (fluvial erosion and transport) associated with a peak,
sediment production decreases over time until a "normal" level of sediment movement is reached. By further developing the
model, Ballantyne (2002a) describes this paraglacial landscape adjustment using the "sediment exhaustion model", which is
based on a hypothetical paraglacial system. Several variable factors determine the duration of this period, such as sediment
release and the rate of sediment reworking. Following the sediment exhaustion model, the rate of sediment reworking of
glacigenic sediments in proglacial areas decreases exponentially if the sediment release rate only depends on sediment
availability (Ballantyne, 2002a, 2002b).
Ballantyne and Benn (1994) and Curry (1999) describe the paraglacial slope adjustment of lateral moraines by analysing the
formation of gully systems on lateral moraines and the corresponding alluvial fans and debris cones (both in western Norway).
These systems result from weathering and erosion, such as fluvial erosion, slope wash, debris flows, smaller slope failures,
and ground/snow avalanches (Ballantyne, 2002a, 2002b; Curry et al., 2006; Haas et al., 2012; Dusik et al., 2019). Material is
deposited in the gullies (e.g. by nival processes, fluvial activity and sidewall collapse) and is then transported downslope,
mainly by debris flows triggered in the gully heads after heavy rainfall or after rapid snowmelt (Ballantyne and Benn, 1994;
Ballantyne, 2002b; Curry et al., 2006). Similarly, large deformations such as deep-seated slope failures and landslides with
low frequency and high magnitude also occur (Mattson and Gardner, 1991; Blair, 1994; Hugenholtz et al., 2008; Altmann et





al., 2020; Cody et al., 2020; Betz-Nutz, 2021; Zhong et al., 2022). These erosion processes are primarily driven by temperature
and precipitation events, which have been subject to change in recent years and decades (Serquet et al., 2011; Brugnara et al.,
2012; Mankin and Diffenbaugh, 2015; Klein et al., 2016; Beniston et al., 2018; Hock et al., 2019; IPCC, 2021; Pepin et al.,
2022). Spring-time snowmelt provide important preparatory steps for sediment transport processes, such as loosening of the
upper layers of sediments of the slope or through the delivery of material into the gulllies by nival processes, which is then
transported downslope by debris flows in the summer months (Haas et al., 2012; Dusik et al., 2019), which is considered as
the most important process occurring (Ballantyne, 2002a; Curry et al., 2006). Dusik (2019) also shows a positive correlation
between the number of mass movements and the number of extreme precipitation intensities, the number of certain threshold
exceedances for extreme daily precipitation totals as well as annual precipitation totals. These processes ultimately lead to the
dissection of the upper parts of the lateral moraines which is, however, limited in time (Curry et al., 2006). Curry et al. (2009)
inferred from morphometric measurements along a chronosequence that gullies increase in depth, width, area, and volume
over time, with width increasing significantly more than depth, resulting in the older ones not being as densely gullied.
Furthermore, it is described that the slope gradient decreases over time, e.g. Ballantyne and Benn (1994) report an average of
5° (in 48 years between 1943 and 1991). Betz-Nutz et al. (2022) document a range of slope gradient changes between -3.2°
and +6.6° between the ~1950s and 2018 (~68 years), showing that both increases and decreases can occur. Ballantyne and
Benn (1994), Curry (1999) and Curry et al. (2006) give average annual erosion rates of different gully systems over several
decades estimated by the volume of the gullies. Curry et al. (2006) showed at different test sites in the Swiss Alps that the
maximum extent of gullies is reached after 50 years of ice release and that sediment filling and stabilisation occurs after 80-
140 years of deglaciation. While 50% of the available sediment is exhausted after 10-50 years, it can take several centuries
until the paraglacial adjustment process is completed (Curry et al., 2006). Schiefer and Gilbert (2007) show, based on
quantitative analyses (via stereo-photogrammetry using historical aerial images), a significant decrease in the geomorphic
activity of gully systems on lateral moraines over several decades and different epochs in the glacier forefield of the Lillooet
Glacier (Canada, British Columbia). Carrivick et al. (2013) generally confirm the concept of paraglacial adjustment by showing
decreasing morphodynamics with increasing distance from the glacier as they have been ice-free for a longer time. However,
the lower morphodynamics observed in the distal areas of the glacier forefields could also be due to the generally lower slope
gradients there (Betz-Nutz et al., 2022). Lane et al. (2017) showed in the glacier forefield of Haut Glacier d'Arolla (Switzerland,
Valais) that there are no indications of filling in the developed gully systems, which indicates that they are still in the incision
phase. Betz-Nutz et al. (2022) show with the use of historical aerial photographs (processed by SfM-photogrammetry) that the
paraglacial adjustment process over decades is very variable. While 13 out of 20 moraine sections showed decreasing erosion
rates over decades, divided into several epochs, six showed almost constant activity and one section even showed a substantial
increase in erosion rate.
The period of paraglacial landscape adjustment is also influenced by upcoming vegetation, which can be considered both a
consequence and a cause of slope stabilisation (Eichel et al., 2016; Haselberger et al., 2021; Haselberger et al., 2022).



Nevertheless, bound solifluction processes can occur under a dense vegetation cover and are therefore not an absolute sign of
stabilisation (Draebing and Eichel, 2017).
The generation of multitemporal accurate and precise digital elevation models (DEMs) and the resulting DEM of Differences
(DoDs) by different remote sensing methods and techniques, which have been established in geomorphological research in
recent years, enabled the detection of changes in the Earth's surface in high spatial and temporal resolution (Pulighe and Fava,
2013; Nebiker et al., 2014; Tarolli, 2014; Smith et al., 2016; Eltner et al., 2016; Sevara et al., 2018; Okyay et al., 2019). By
processing overlapping high-resolution digitised historical aerial images (HAI) of high alpine geosystems, using SfM-MVS
(Structure-from Motion with Multi-View-Stereo) digital stereo-photogrammetry in combination with current airborne LiDAR
(Light Detection And Ranging) data into DEMs and the corresponding DoDs, landscape changes in these areas can be
reconstructed over several decades (Midgley and Tonkin, 2017; Mölg and Bolch, 2017; Lane et al., 2017; Betz et al., 2019;
Altmann et al., 2020; Fleischer et al., 2021; Betz-Nutz, 2021; Stark et al., 2022; Piermattei et al., 2022). The spatial distribution
of positive and negative DoD elevation changes enable various analyses, such as the reconstruction and interpretation of
individual geomorphological processes (Dusik, 2019) or the calculation of morphological budgets (Altmann et al., 2020).
Furthermore, by applying flow routing algorithms and the accumulation of DoD values accordingly, sediment yield (SY) from
the contributing area of each cell can be determined: Pelletier and Orem (2014) used repeat airborne LiDAR-based DEMs
before and after a wildfire and calculated for each pixel the net sediment volume exported by geomorphological processes.
Further applications of this methodology have been published by Wester et al. (2014), who calculated the total SY by applying
a weighted flow accumulation algorithm, and Heckmann and Vericat (2018), who further developed the approach by
calculating a spatially distributed measure of functional sediment connectivity on a proglacial slope. Neugirg et al. (2015a;
2015b; 2016) showed a positive correlation between log SY (calculated by accumulated DoD values on slopes) and the
corresponding log SCA (sediment contributing area), respectively log CA (catchment area), both extracted at randomly
selected cells of the channel network (so-called "virtual sediment traps", VST). Besides to these studies, which were carried
out on hillslopes in the Northern Alps (Germany, Lainbach valley and Arzbach valley) and at a former iron ore mine on the
island of Elba in the Tyrrhenian Sea (Italy, next to Rio Marina)), this approach was also applied to one proglacial slope in the
Kaunertal (Austria, Tyrol) by Dusik (2019) and Dusik et al. (2019). One advantage of this approach is that it can be used to
determine not only the size of SY (which can be compared with previous epochs, for example), but also the variability of SY
in the AOI within an epoch (spatial pattern of SY within the AOI), which is not possible, for example, when calculating simple
erosion rates, where only the volume of the total change can be computed.
In order to better understand the paraglacial adjustment process of lateral moraines, we continue the application of the approach
of Neugirg et al. (2015a; 2015b; 2016), Dusik (2019) and Dusik et al. (2019) to different LIA lateral moraines in the central
Eastern Alps in this study, which has so far only been carried out on one proglacial slope and over a short epoch of a few
months (Dusik et al., 2019). Secondly, we show volume calculations of the entire AOIs to determine the total sediment yield
(and erosion rates). Combining high-resolution historical and current DEMs and the corresponding DoDs, we show, the
quantification and analysis of gully system morphodynamics at 12 different sections in the upper reaches of lateral moraines




in five different glacier forefields over a total epoch of several decades (1953-2019) with several survey periods (~1950s to
~1970s, ~1970s to ~2000s and ~2000s to 2017/2019). By using simulated climate data of the glacier forefields we were able
to investigate, besides system-internal influences, also external impacts on the morphodynamics, which have not been
considered in long-term studies on erosion of LIA lateral moraines so far.
**2 Study Area**
The areas of interest (AOIs) are located in different high alpine geosystems along a north-south axis in the central Eastern Alps
and are situated north (Horlachtal and upper Kaunertal) and south (upper Martelltal) of the main alpine divide. In these valleys,
the AOIs are located within five glacier forefields on lateral moraines formed by the glaciers during their maximum glacier
extent during the LIA around 1850 (Figure 1). The Horlachtal is located in the Stubai Alps (Tyrol, Austria), which is a tributary
of the Oetztal (Geitner, 1999; Rieger, 1999). The investigated section of the Horlachtal is located in the side valley and sub-
catchment Grastal (glacier forefield Grastalferner), which is oriented in a north-south direction. Geologically, the Horlachtal
is located in the Oetztal Massif, where gneisses and mica schists dominate (Becht, 1995; Geitner, 1999). The Kaunertal is also
located in the Oetztal Alps (Tyrol, Austria) and is oriented in a north-south direction. This valley geologically belongs to the
Austroalpine crystalline complex (Tollmann, 1977; Geological Survey of Austria, 1999) where crystalline rocks, mainly ortho-
and paragneisses, dominate (Vehling, 2016). The AOIs within the Kaunertal are located in the glacier forefields of the
Gepatschferner, another glacier outlet of the Gepatschferner, the so-called Münchner Abfahrt (MA), and the Weißseeferner.
The Martelltal is a southwest-northeast oriented valley located in the Ortler-Cevedale group (South Tyrol, Italy) and belongs
geologically to the Ortler-Campo Crystalline, where quartz phyllite dominates with layers of e.g. shales, gneisses and marbles
(Mair and Purtscheller, 1996; Staindl, 2000; Mair et al., 2007). The two AOIs are located in the glacier forefield of the
Hohenferner. All valleys are characterized by the continental climate and low annual precipitation sums of the inner alpine dry
region (Becht, 1995; Hagg and Becht, 2000; Veit, 2002; Hilger, 2017; Betz-Nutz, 2021). The AOIs are characterized by very
low vegetation cover, intense paraglacial morphodynamics and typical unsorted moraine material. Table 1 and Figure 1 give
an overview of the location as well as the characteristics of the AOIs.
**Table 1: Characteristics of the AOIs. Values were derived from 2017 DEM (Kaunertal) and 2019 DEM (Horlachtal and Martelltal).**

| AOI | Location (Centre) (ETRS89/ UTM Zone 32N, EPSG Code: 25832) | Elevation (Ellipsoidal heights) (m) | Aspect | Size (m²) | Max. length of delinated AOI (downslope) (m) | Mean (and max.) slope gradient (°) | At least ice-free since (years)* | Glacial or dead ice influence at the foot of the slope |
|---|---|---|---|---|---|---|---|---|
| HG1 | E 652032, N 5218283 | 2659-2696 | W | 1647 | 43 | 37.9 (46.8) | 1860 (159) | Not detectable |
| KG1 | E 632991, N 5193590 | 2183-2262 | W | 12431 | 124 | 41.5 (69.3) | 1937 (80) | Not detectable |
| KG2 | E 633140, N 5193339 | 2244-2321 | SW | 8814 | 59 | 43.8 (61) | 1933 (84) | Until 2006 |
| KG3 | E 633421, N 5193204 | 2329-2400 | S | 3123 | 29 | 38.5 (48.3) | 1872 (145) | Not detectable |
| KG4 | E 634596, N 5193101 | 2540-2620 | SW | 6193 | 99 | 41.1 (61.3) | 1929 (88) | until today |





| KG5 | E 634789, N 5192997 | 2580-2645 | SW | 3531 | 77 | 44.3 (57.1) | 1913 (104) | until today |
| KM1 | E 632904, N 5192058 | 2443-2486 | E | 2025 | 23 | 39.8 (46.9) | 1903 (114) | Not detectable |
| KM2 | E 632783, N 5191632 | 2560-2598 | E | 2534 | 30 | 45.7 (56.7) | 1901 (116) | Until 2006 |
| KW1 | E 631025, N 5192561 | 2546-2603 | SW | 2951 | 38 | 41.6 (54.4) | 1924 (93) | Not detectable |
| KW2 | E 631204, N 5192213 | 2682-2714 | SW | 3638 | 49 | 39.9 (53.4) | 1937 (80) | Until 2006 |
| MH1 | E 628937, N 5147454 | 2704-2729 | E | 1475 | 26 | 35.5 (51.6) | 1921 (98) | Not detectable |
| MH2 | E 629426, N 5147413 | 2755-2796 | SW | 3983 | 45 | 45.3 (72) | 1943 (76) | Until 2004/2005 |

**\*Determination of complete deglacialisation is based on an interpolation between the two glacier extensions within which the AOIs have become ice-**
**free by calculating the euclidean distance as proposed by Betz-Nutz et al. (2022).**





**Figure 1: Location of AOIs, glacier extents (Sources in Table 2) and location for meteorological data extraction (for corresponding analysis, see sec. 3.3). Large-scale elevation data (DSM, 25 m) (centre right) are based on SRTM and ASTER GDEM (Copernicus, 2016). DEMs (1 m) (right and bottom right) are based on airborne LiDAR (ALS) data from 2017 (Kaunertal) and 2019 (Horlachtal and Martelltal) (see sect. 3.1.1). Orthophotos (from 2020) are provided by the Province of Tyrol (Horlachtal and Kaunertal) and by**



**the Autonomous Province of Bolzano, South Tyrol (Martelltal). The glacier extent of Groß and Patzelt (2015) is based on mapping**
**of the LIA lateral moraines and field surveys based on orthophotos. In the process of this study, these mappings were slightly**
**modified so that they fit to the maximum glacier extent (LIA lateral moraines) more accurately. The glacier extents end of LIA,**
**1918, 1945 and 1959 in the Martelltal have already been described by Betz et al. (2019).**
**Table 2: Sources of the glacier extents.**

| Valley | Year | Source |
|---|---|---|
| Horlachtal | End of LIA | Groß and Patzelt (2015) |
| | 1889 | Gedächtnisspeicher Ötztal (Austria, Längenfeld), K&K Militärgeographisches Institutsarchiv* |
| Kaunertal | End of LIA | Groß and Patzelt (2015) |
| | 1886/1887 | Finsterwalder and Schunck (1888)* |
| | 1922 | Finsterwalder (1928)* |
| | 1953 | Images of BEV, DoD 1953/2017*** |
| | 1970/1971 | Images of the Office of the Tyrolean Government, DoD 1970/1971-2017*** |
| | 2006 | Province of Tyrol, DoD 2006-2017*** |
| | 2017 | Chair of Physical Geography, Cath. University Eichstätt-Ingolstadt, SEHAG-project (See sect. 3) ** |
| | 2020 | Province of Tyrol, orthofoto** |
| Martelltal | End of LIA | Mapped on base of visible moraines and descriptions of Finsterwalder (1890) |
| | 1918 | Spezialkarte 1:75.000 of BEV* |
| | 1945 | Images of the IGMI, orthofoto* |
| | 1959 | Images of IGMI, DoD 1959-2019*** |
| | 2004/2005 | Autonomous Province of Bolzano, DoD 2004/2005-2019*** |

***based on historical map, **based on orthophoto and/or hillshade and ***based on DoD (SfM-MVS/photogrammetry and/or ALS).**
**3 Material and Methods**
**3.1 Generation of the topographic data**
**3.1.1 Processing of airborne LiDAR and photogrammetric/SfM-MVS point clouds**
Several data sets were used for the reconstruction of the terrain surface for the entire catchments. These include both current
airborne LiDAR data and historical aerial image series (Figure 2). Thus, the epochs are based on the availability and quality
of the data.



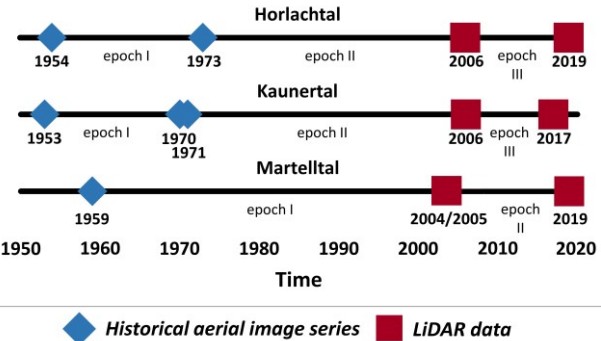

**Figure 2: Data and epochs.**

To determine the recent morphodynamics in the respective AOIs, available airborne LiDAR data from 2004/2005 to 2019 were used. The 2004/2005 and 2006 data of the three valleys were provided by the Autonomous Province of Bolzano and the Province of Tyrol (Table 3). The latest ALS datasets of each valley (2017 and 2019) were collected in own ALS flight campaigns of the Chair of Physical Geography at the Catholic University of Eichstätt-Ingolstadt (Table 3) (Stark et al., 2022). In this case, LiDAR data sets were collected using previously determined flight strips. Direct georeferencing (position and altitude) of the trajectories was determined by Global Navigation Satellite System (GNSS) rover antenna and an Inertial Measurement Unit (IMU) (Applanix AP 20), both located in the laser scanner. In addition, GNSS correction data were acquired on the ground during the flight missions using a dGNSS antenna (Figure 3). Subsequently, the GNSS/IMU trajectory data were processed in three steps. This included, (i) the calculation of precise trajectories using the software PosPac MMS (Applanix), (ii) the attachment of raw scans to the flight lines using the software package Riegl RiProcess, and finally (iii) a strip adjustment in the processing software OPALS (Pfeifer et al., 2014) using the approach of Glira et al. (2015).

**Table 3: Overview of the ALS (and DEM) data.**

| Valley | Date of acqui-sition | Source/Purpose | Laser-scanner | Field of view (°) | Flying altitude (metre above ground) | Air-speed (kn) | Laser pulse Measuring frequency (khz) | Wave-length (nm) | DEM res. or mean point density of the AOIs (points/m²) |
|---|---|---|---|---|---|---|---|---|---|
| Horlachtal | 05.09.2006 | Province of Tyrol | N/A | N/A | N/A | N/A | N/A | N/A | DEM, 1 m |
| | 08.08.2019 | SEHAG project ("SEnsitivity of High Alpine Geosystems to climate change since 1850") | Mobile laser scanner VP1 (Riegl VuxSys-LR) | 180 | ~150 | ~45 | 200 | 1550 | 24.1 |
| Kaunertal | 05.09.2006 | Province of Tyrol | N/A | N/A | N/A | N/A | N/A | 999 | 3.4 |
| | 05.07.2017 | PROSA project ("High-resolution measurements of morphodynamics in rapidly changing | Mobile laser scanner VP1 (Riegl | 180 | ~150 | ~45 | 200 | 1550 | 35.7 |



| | | PROglacial Systems of the Alps") | VuxSys-LR) | | | | | | |
|---|---|---|---|---|---|---|---|---|---|
| Martelltal | 2004/2005 | Autonomous province of Bolzano | N/A | N/A | N/A | N/A | N/A | N/A | 1.4 |
| | 09.08.2019 | SEHAG project ("SEnsitivity of High Alpine Geosystems to climate change since 1850") | Mobile laser scanner VP1 (Riegl VuxSys-LR) | 180 | ~150 | ~45 | 200 | 1550 | 13.3 |

189

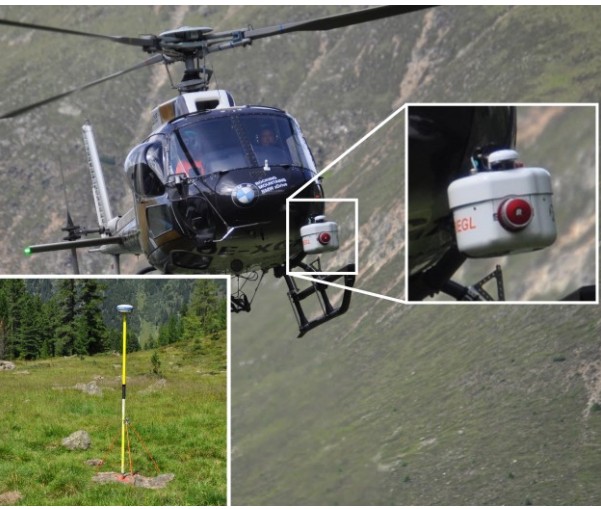

190

**Figure 3: ALS data collection on 08.08.2019 in Horlachtal. Helicopter with nose-mounted VP1 laser scanner as well as the ground station which recorded the dGNSS raw data during the flight time (Stonex S9III).**

In order to extend the temporal scope of this study by several decades (until 1953), previously digitised (high-resolution) overlapping HAI were processed into historical DEMs. Except for the 1959 Martelltal-survey, camera distortion parameters and focal lengths were provided for all data with the respective camera calibration certificates (Table 4).

The digitised image series were processed with the Agisoft Metashape Professional software package (Version 1.6.6; Agisoft LLC) using Structure from Motion (SfM) photogrammetry with multi-view-stereo (MVS) algorithms to generate high-resolution point clouds. The generation of point clouds from digitised historical (aerial)image series requires different preparation and processing steps. First, all images of each series were resized to a common image size (uniform number of pixels along the x- and y-axis) without changing the image content. This step was necessary so that the software can assign all images to the same camera (source) and was carried out using Adobe Photoshop (CS6). This is of enormous importance in order to be able to use the appropriate distortion parameters for the respective camera models for the calculation. After, the image sets were imported into single folders and a common global coordinate system (ETRS89/UTM zone 32N; EPSG code: 25832) was defined. Next, all images were masked to exclude the black borders/frame (instrument stripes with the camera metadata) in order to avoid interference with the orientation of the cameras (Gomez et al., 2015). Before the initial processing



of images we defined the fiducial mark information and lens distortion parameters in order to set the metric dimension of
images and lenses. This informations were included and used for the alignment of single images (SfM).
Since a global exterior orientation requires a large number of precisely surveyed ground control points (GCPs) distributed
throughout the area, we used highly-precise ALS datasets with millimetre accuracy (2019 Horlachtal; 2017 Kaunteral) to
extract these GCPs and to define the exterior orientation of all data. The selection and extraction of GCPs was based on clearly
identifiable objects (e.g. rock formations) that were also considered as stable (geomorphologically unchanged) over the entire
observation period. If a calibration certificate was available, the film camera option was used, fiducial marks defined, the focal
length set and fixed. All other lens distortion parameters ($C_x$, $C_y$, $k_1$, $k_1$, $k_1$, $p_1$ & $p_2$) were estimated and adjusted fully automatic
using the auto-calibration function. In case of missing camera calibration certificate, an auto-calibration (no film camera) was
performed. Both options were proposed by Stark et al. (2022).
According to these pre-processing steps, the point clouds were generated by (i) initial joint orientation of the images, (ii)
selection of ground control points (GCPs), (iii) final camera orientation (bundle block adjustment) including scale definition,
and (iv) calculation of dense point clouds.
The processing of the 1959 point cloud, which was used in this study, is already described in Betz et al. (2019).
**Table 4: Overview of acquired historical image series for point cloud generation and corresponding DEMs by photogrammetry/SfM.**

| | 1953 (Kaunertal) | 1954 (Horlachtal) | 1959 (Martelltal) | 1970 (Kaunertal) | 1971 (Kaunertal) | 1973 (Horlachtal) |
|---|---|---|---|---|---|---|
| Source/ Purpose | BEV/Forest condition estimation; Flight C | BEV/Forest condition estimation; Flight D | IGMI | Office of the Tyrolean Government/ Tyrolean state Surveying flight | Office of the Tyrolean Government/ Tyrolean state Surveying flight | Office of the Tyrolean Government/ Tyrolean state Surveying flight |
| Date of acquisition | 31.08.1953/ 01.09.1953/ 08.09.1953 | 31.08.1954/ 04.09.1954 | 09.09.1959/ 20.09.1959 | 29.09.1970 | 18.08.1971 | 06.08.1973 |
| Flying altitude (m a.s.l) | ca. 5955/ unknown ca. 5850 | ca. 6110/ ca. 5920 | ca. 5100/ ca. 5000 | ca. 8665 | ca. 5025 | ca. 4900 |
| Camera | Wild RC/5 | Wild RC/5 | Santoni | Wild RC5/RC8 | Wild RC5/RC8 | Wild RC5/RC8 |
| Number of images | 36/51/63 | 32/4 | 2/6 | 26 | 31 | 88 |
| Focal length (mm) | 210.11 | 210.23 | 153.41 | 210.43 | 209.48 | 210.43 |
| Scanning Resolution (µm) | 15 | 15 | N/A | 12 | 12 | 12 |
| Format | TIFF | TIFF | TIFF | TIFF | TIFF | TIFF |
| Calibration protocol available | yes | yes | no | yes | yes | yes |
| Number of GCPs | 100 | 74 | 23 | 88 | 29 | 67 |



| Mean point density (points/m²)* | 8.5 | 3.7 | 4.9 | 13.3 | 15.7 | 20.5 |
| Ground resolution (cm/pix) ** | 22.5 | 34.8 | 19.6 | 19 | 17 | 13.8 |
| RMS reprojection error (pix) ** | 0.48 | 0.51 | 1.55 | 0.86 | 0.44 | 0.45 |

**\*refers to the exact AOIs, \*\*refers to the entire data set**
**3.1.2 Digital elevation model (DEM) and DEM of Difference (DoD) processing**
Although all point clouds were finally available in the same coordinate system (ETRS89/UTM Zone 32N, EPSG Code: 25832), a
local adjustment of each AOI was carried out to obtain the highest possible accuracy of the subsequent DoDs to be calculated.
For this purpose, stable areas, i.e. geomorphologically unchanged areas such as rock outcrops or stable areas on the lateral
moraines, were mapped next to each AOI based on orthophotos. To match the point clouds as well as possible, the Iterative
Closest Point algorithm (ICP) (Besl and McKay, 1992; Bakker and Lane, 2017) implemented in SAGA-LIS (Conrad et al.,
2015) was used for fine registration. Previously, the LiDAR based point clouds were further processed in the software SAGA-
LIS (LIS Pro 3D) from Laserdata (laserdata.at) in combination with Python and R to prepare point clouds for the generation
of high-resolution digital elevation models (DEMs). This included the removal of outliers (Remove Isolated Points), a ground
classification (to remove vegetation), which was carried out with a modified approach according to Hilger (2017) and the
achievement of more homogeneous point clouds with the tool 3D Block Thinning (PC) in SAGA-LIS. The point clouds were
then converted into DEMs using the Point Cloud to Grid tool in SAGA-LIS (elevations of points averaged for each raster cell;
cell sizes for Horlachtal and Kaunertal 1m, for Martelltal 2m). Finally, the DoDs were generated by subtracting the individual
DEMs from each other to determine the positive and negative elevation changes of the earth's surface.
**3.1.3 Uncertainty assessment**
The presence of various uncertainties in differently generated DEMs (Hodgson and Bresnahan, 2004; Bakker and Lane, 2017)
also leads to uncertainties in the resulting DoDs (Lane et al., 2003; Rolstad et al., 2009; Cavalli et al., 2017; Anderson, 2019).
Therefore, an uncertainty assessment was carried out using the DoD values from stable areas near each AOI. The size of the
stable areas varied between 25% and 75% of the size of the corresponding AOI. In addition to the estimation of the precision
(Std Dev) and accuracy (RMSE), the arithmetic mean, minimum and maximum values were also determined (Figure 4).






**Figure 4: Uncertainty assessment.**



To determine the uncertainty of the sediment volume change (total sediment output, Figure 7), the error propagation method
for uncorrelated, correlated and systematic error according to Anderson (2019) was applied. We chose not to threshold our
DoDs by a Level of Detection following Anderson's (2019) clear recommendation not to apply thresholding to net volumetric
change analysis where thresholding can lead to biased results.
For the final determination of the total error, the following formula was applied Eq. (1):
$\sigma_v = \sqrt{\sigma_{v,re}^2 + \sigma_{v,sc}^2 + \sigma_{v,sys}^2},$ (1)
where $\sigma_{v,re}$ is the uncorrelated error, $\sigma_{v,sc}$ spatially correlated error and $\sigma_{v,sys}$ systematic error.

**3.2 Derivation of the regression lines**

In this study, we followed the SCA approach of Neugirg et al. (2015a; 2015b; 2016) and Dusik et al. (2019), who applied this
approach at the slope scale and replaced real sediment traps in the channels, as originally based on the work of Haas (2008)
and Haas et al. (2011), with so-called virtual sediment traps (VSTs) in modelled channels in a DEM. The SCA model represents
a set of simple DEM-based rules according to Heinimann et al. (1998) for delineating those geomorphologically active areas
that potentially deliver sediment to the channel network (and hence constitute the sediment contributing area of the latter).
This approach is similar to the "effective catchment area" proposed by Fryirs et al. (2007) and Fryirs (2013). By selecting
different parameters related to topography and landcover information, namely the minimum channel gradient threshold (for
longitudinal (de-)coupling), the minimum slope gradient threshold (for lateral (de-)coupling), the maximum distance from
channel (slope length) and a weighting of the vegetation cover (representing the role impedance of vegetation as a disturbing
factor into sediment transport), Haas (2008) and Haas et al. (2011) reduced the hydrological catchment accordingly to the
sediment supplying and thus the sediment contributing area (SCA). A correlation between the size of the SCA, which thus
corresponds to a subset of the hydrological catchment, and the computed sediment yield (determined by sediment traps in the
channels) could be shown, but no correlation between the size of the hydrological catchment and the sediment yield. This
shows that only a certain part within a hydrological catchment is geomorphologically active, providing sediment to the
channels and subsequently transporting it downstream, as covered areas and areas with low gradients (hillslope and channel
sections) reduce sediment connectivity within a catchment. Linear regression analysis was used to show this significant
correlation, which is formulated as Eq. (2):
$y = \text{intercept} + \text{slope} * x,$ (2)
where $y$ is (log.) mean annual bedload sediment yield and x (log.) SCA.
The SCA model uses an empirical relationship between log. sediment contributing area as the independent variable and log.
mean annual bedload sediment yield as the dependent variable. Thus, the SCA can be used as a predictor of sediment delivery
in alpine catchments. This has already been confirmed in several studies in both small and large catchments (ranging from



hectare to square kilometres) and in different regions such as the Northern Calcareous Alps (Haas, 2008; Haas et al., 2011;
Sass et al., 2012; Huber et al., 2015) and the French Northern Alps/Prealps (Altmann et al., 2021).
Finally it can be stated that a linear dependency of two variables x and y on a log-log-scale has a fundamentally different
behavior than a usual linear dependency. In our case, we have y = log(SY) and x = log(SCA). Back-transformation of Eq. (2)
using the *exp* function yields gives the following relation between SCA and SY, Eq. (3):
$SY = \exp(\text{intercept}) * SCA^{\text{slope}}$ (3)
Thus, the relation between SY and SCA is a polynomial of the form $y = a * x^b$. In particular, the slope in the log-log model
represents the exponent of the polynomial in the standard model. The relation between SY and SCA is (nearly) linear if slope
is (close to) one. In this case, the exponential of the intercept in the log-log model represents the slope of the linear relation in
the standard model, meaning that independent of the actual size of the SCA, one square meter provides the same amount of
SY, given by $exp(intercept)$. On the other hand, if the slope in the log-log model is considerably greater than one, the
standard model shows a polynomial behaviour, meaning that in the same AOI, increasing the SCA provides more SY per
square meter.
The steps of the SCA approach of this study are composed as follows and were implemented in SAGA LIS and R. The elevation
changes in DoDs (using no threshold) generated from multitemporal data were routed downslope and accumulated using the
D8 algorithm (O'Callaghan and Mark, 1984). The resulting accumulated DoD values (accDoD) in every raster cell corresponds
to the net volume of the sediment balance within its contributing area. On steep slopes, accDoD will be negative and represents
the sediment yield of this contributing area (Pelletier and Orem, 2014); if it is close to zero, it means that all eroded sediment
has been re-deposited within the contributing area. As in the previous SCA studies by Neugirg et al. (2015a; 2015b; 2016), the
application of the parameters used in the original SCA model (Haas, 2008; Haas et al., 2011), which lead to the reduction of
the hydrological catchment to the SCA, is omitted because the AOIs and the modelled channels are consistently steep,
uncovered and have short slope lengths, which makes this reduction obsolete. Therefore, the SCA is identical to the catchment
area (CA) in this study.
In detail, channel initiation points were delineated using a threshold of 20 m² of the flow accumulation that was computed
using the D8 algorithm (O'Callaghan and Mark, 1984). Channels that were shorter than 10 m were discarded. To ensure
statistical independence through avoiding overlapping contributing areas, a stratified sampling scheme was adopted that
included one randomly selected raster cell per channel. Pairs of values (SY and the corresponding SCA size) were randomly
extracted from the corresponding channels (representing the VSTs) for each AOI and a regression line were calculated
accordingly. To quantify the uncertainty due to random selection, this sample was repeated 100 times, resulting in 100
regression models of SY on SCA.
Furthermore, we added two conditions and further developed the SCA approach accordingly. In order to obtain more stable
regression lines, the range of values of the SCA size was divided into quartiles (with equal number of cells within the quartiles)
to ensure a homogeneous distribution of the extracted values. Additionally samples that contained points with a high leverage





(greater than 0.5) in the regression model were discarded, and the sampling was repeated until a number of 100 samples was
reached.



**Figure 5: Example derivation of the statistical relationship at AOI KG2.**
**3.3 Calculation of the sediment output**
Additionally, the total sediment output volume, the mean annual sediment output (divided by the corresponding number of
years) and the specific mean annual sediment output (additionally divided by the area of the AOI) were calculated for each
AOI and epoch.
The following equation was used for this (4):
$V = \sum D \, oD * L^2,$ (4)
where $\Sigma$ DoD is the sum of the corresponding DoD subset values and $L^2$ is the cell size.
**3.4 Generation of meteorological data**
Using data generated with a regional climate model (RCM), the influence of the changes in climate forcing (air temperature
and precipitation) on morphodynamics was investigated. For dynamical downscaling of climate data for the  beginning of the
study period until 2015, we used the Advanced Research Version of the Weather Research and Forecasting (ARW-WRF)
model (version 4.3), which is based on fully compressible and non-hydrostatic equations (Skamarock and Klemp, 2008). The
20th Century Reanalysis version 3 (20CRv3) dataset (Compo et al., 2011; Giese et al., 2016; Slivinski et al., 2019), with a
spatial and temporal resolution of 1°x1° and three hours, respectively, was used as driving data (initial and boundary
conditions). The simulation was performed in three nested domains with grid spacing of 18- (Domain 1), 6- (Domain 2), and
2-km (Domain 3). For our simulations, we mainly used the physics and dynamics options proposed by Collier and Mölg
(2020), and are listed in Table 5. However, the Noah land surface model, prescribed eta levels by Collier et al. (2019), and the
24 United States Geological Survey (USGS) land use categories were used. The temporal resolution of simulated data in D3
is 1 hour for temperature and 15 minutes for precipitation.
**Table 5: Overview of the WRF configuration.**

| Domain configuration | |
| --- | --- |
| Horizontal grid spacing | 18-, 6-, 2-km (D1, D2 and D3) |
| Grid dimensions | 190 x 190, 151 x 142, 121 x 139 |
| Lateral boundary condition | variable (20CRv3 at 1°x1°, 3-hour) |
| Time step | 90, 30, 10 s |
| Vertical levels | 50 |
| Model top pressure | 10hPa |
| | |
| Model physics | |
| Microphysics | Morrison (Morrison et al., 2009) |
| Cumulus | Kain-Fritsch (none in D3) (Kain, 2004) |
| Radiation | RRTMG (Iacono et al., 2008) |



| Planetary boundary layer | Yonsei State University (Hong et al., 2006) |
| Atmospheric surface layer | Monin Obukhov (Jiménez et al., 2012) |
| Land surface | Noah (Chen and Dudhia, 2001) |
| | |
| Dynamics | |
| Top boundary conditions | Rayleigh damping |
| Diffusion | Calculated in physical space |


For the period from 2016 to the end of the study period (2017/2019), the ERA5 reanalysis dataset (Hersbach et al., 2018) was
used (spatial resolution: 55 km, temporal resolution: 1 hour). The different meteorological datasets were combined and divided
into the corresponding study epochs.For this purpose the temporal resolution of the precipitation data simulated with WRF
was adjusted to one hour to fit the ERA5 temporal resolution. The simulated temperature and precipitation data were extracted
at the location of each of the five glacier forefield (Figure 1). These are the centres of the respective AOIs and represent the
corresponding glacier forefield. In addition, a corresponding elevation correction of the climate data was applied for
temperature.
For the analysis, we used the mean annual air temperature (2 metres above ground), as well as the corresponding trends and
the mean number of ice days (days with maximum temperature <0°C). In addition, the number of warm air inflows from
October to May was determined in order to identify corresponding snowmelt processes on the AOIs. A warm air inflow is
defined as a period of at least 3 days in which more than 70% is above 0°C, following a previously colder period of 5 days
(100% below 0°C). In addition, the precipitation patterns were analysed. For this purpose, the mean annual precipitation totals,
the mean annual winter (October to May) and the mean annual summer precipitation totals (June to September) as well as the
corresponding trends were determined in order to identify seasonal changes. Furthermore, various continuing classes (4 mm
for one-hour resolution and 10 mm classes for daily totals) were used to analyse corresponding changes in individual extreme
events and daily precipitation totals. Individual precipitation events were defined as one event, regardless of length, if they
were contiguous throughout, and were separated if there was no precipitation for at least one hour. To minimise the noise
generated in the data, both datasets were also filtered for extremely small events by changing the values from <0.01 mm to 0
mm. The calculation of the mean annual winter precipitation was always carried out over the entire winter.For example, for
the winter of 1953, data from October 1952 to May 1953 was included. The average summer precipitation was calculated
accordingly from June 1953 to September 1953. Furthermore, precipitation was differentiated into snow and rain events. The
determination of a threshold to distinguish rain from snowfall is very dynamic in mountainous regions and difficult to estimate.
However, the difference between rain to snow depends mainly on surface air temperature as well as air humidity, with snow
occurring mainly between 0 and 3°C (Froidurot et al., 2014) and the lower the humidity, the higher the probability of snowfall
is. In this study, the threshold from rain to snow was defined at ≤0°C, as below this temperature rain is almost excluded
(Froidurot et al., 2014; Fehlmann et al., 2018).





## 4 Results

### 4.1 Sediment-Contributing-Area (SCA) approach

All determined regression lines show a positive correlation between log mean annual sediment yield (SY) and log sediment contributing area (SCA) (Figure 6, Appendix A), which means that SY increases with the corresponding SCA. In the following, only the median of the 100 regression lines (median slope) of all AOIs and epochs is used to qualitatively describe corresponding differences. Mostly, there is a decrease in SY between the different epochs and a decrease of the slopes of the regression lines, which is due to a decrease in SY per square metre of the AOIs. With regard to section 3.3, a decreasing intercept together with an almost constant slope close to one over the different epochs (in the log-log-model) (although with a slightly decreasing slope) indicates that the relation between SCA and SY stays nearly constant. The AOI's KG3, KG4, KM1, KM2 and KW2 show such a behaviour. On the other hand, the areas KG1, KG2, KG5, KW1 and MH2 show clearly larger differences in the epochs. In the earliest epoch, the slopes considerably larger than one (in the log-log model) show polynomial behaviour, which means that in the same AOI an increasing SCA provides clearly more SY per square meter. In the later epochs, the slopes also tend towards one, so that the models of the different groups become similar. In addition to this general trend (ten AOIs), an increase in SY and an increase in the slope of the regression line for AOI HG1 were observed, showing an increase in sediment dynamics over the epochs in this case, which is in contrast to the previous observations. AOI MH1 shows a similar level of SY (between the epochs) with higher slopes of the regression lines, also indicating an increase in SY. Furthermore, slopes of the regression lines below 1 occur in all epochs, but especially in the second and third.





**Figure 6: Relationships between log SCA and log SY yield for 100 samples of each AOI and the corresponding epochs. In addition, the median regression line (median slope) is represented by a slightly thicker and darker line.**

## 4.2 Volume calculations of the sediment output

The analyses of the sediment output of the AOIs confirm the results of the SCA approach (Figure 7). In general, there is a clear and continuous decrease in mean annual SY of ten AOIs over the different epochs. In contrast to this trend, AOI HG1 shows a clear increase in mean annual SY. AOI MH1 also shows an increase, but at a very low level, which can also be described as a geomorphic activity of a similar level. In total, the mean annual sediment output decreases across the different epochs. Nevertheless, there is also very high temporal and spatial variability of this change on the AOIs HG1 and MH1, which also shows a clear increase in geomorphic activity as well as a slight increase (respectively activity at the same level).



385

**Figure 7. Bar plots of total sediment output (with error range according to Anderson (2019), see sect. 3.1.3 and 3.4), mean annual sediment output and mean annual specific sediment output (erosion rate) of each AOI and epoch.**



## 4.2 Meteorological regime

### 4.2.1 Air temperature

The mean annual air temperature (2 m above ground) of all selected positions of the glacier forefields shows a statistically significant warming trend over the entire study period of 60, 64 and 65 years (Figure 8). Overall, there is a positive total change of +1.75°C (annual trend 0.03; p-value <0.05; $R^2$ 0.39) for the Horlachtal/Grastalferner glacier forefield, +1.68°C (annual trend 0.03; p-value <0.05; $R^2$ 0. 38) for the Kaunertal/Gepatschferner glacier forefield, +1.70°C (annual trend 0.03; p-value <0.05; $R^2$ 0. 38) for the Kaunertal/Gepatschferner Münchner Abfahrt glacier forefield, for the Kaunertal Weißseeferner glacier forefield of +1.64°C (annual trend 0.03; p-value <0.05; $R^2$ 0.36) and for the Martelltal Hohenferner glacier forefield of +2.23°C (annual trend 0.04; p-value <0.05; $R^2$ 0.45).

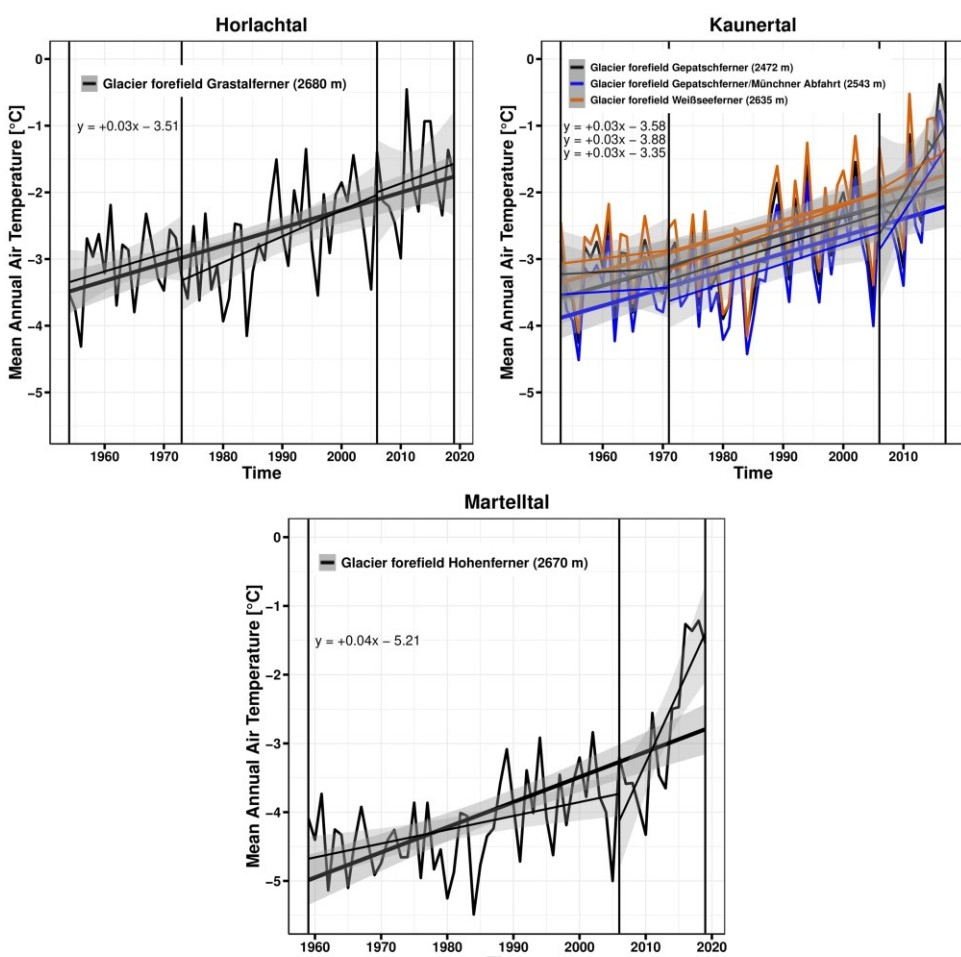

**Figure 8: Mean annual 2 meter air temperature of the glacier forefields within the study epochs (95% confidence interval is included).**



The analysis of the mean annual ice days shows a decrease between the epochs, especially from the second to the third epoch
(in the Martelltal from the first to the second), with a decrease in ice days between 18.4 and 20.7 days, which corresponds to
almost three weeks (Figure 9). The analysis of the mean annual warm air inflows shows a decrease from the first to the second
epoch in the glacier forefields of Grastalferner, Gepatschferner and Gepatschferner/MA, and a more pronounced increase from
the second to the third epoch. In the glacier forefield of Weißseeferner ,there is a consistent increase, the latter being equally
more pronounced, whereas in Martelltal there is only a slight increase. Thus, the number of warm air inflows has generally
increased.

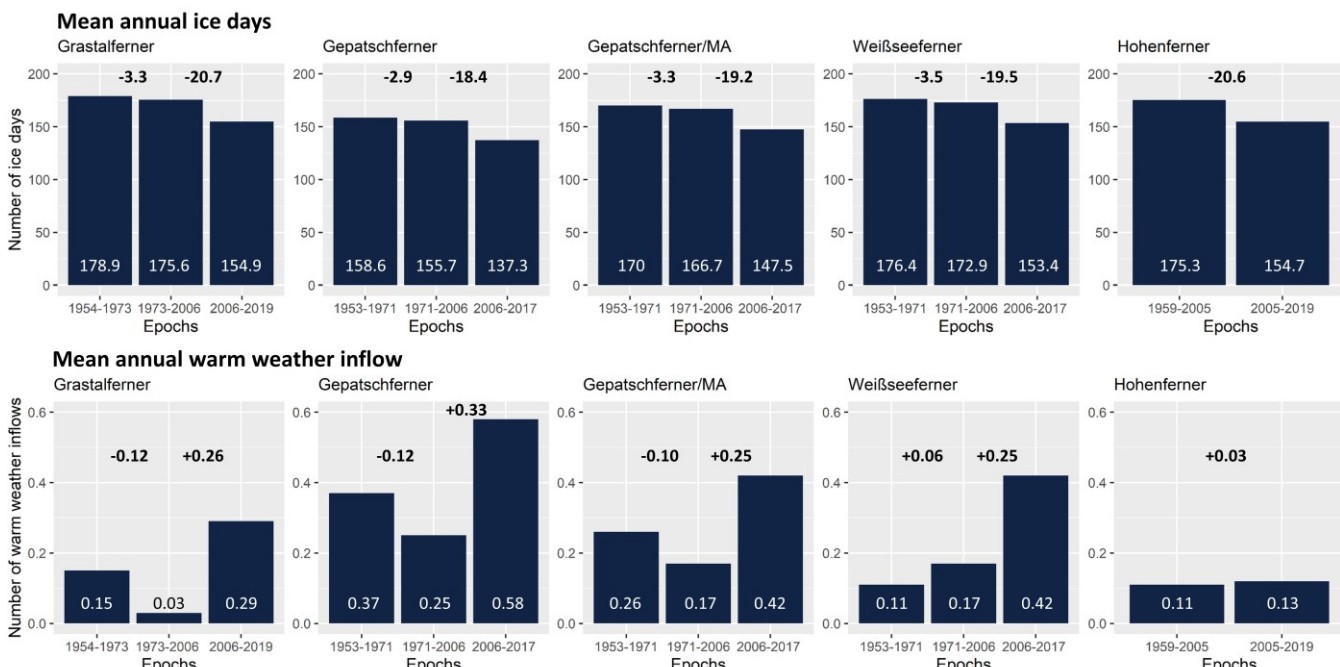

**Figure 9: Mean annual ice days and mean annual warm weather inflows with the corresponding changes between epochs.**
**4.2.2 Precipitation**
Over the entire epochs (60, 64 and 65 years), all study areas show a decreasing trend in mean annual, mean summer and mean
winter precipitation (with the exception of summer precipitation in the Martelltal, which shows a positive trend) (Figure 10).
However, only the changes in winter precipitation (entire study period), summer precipitation (second epoch: 2004/2005-2019)
in the Martelltal (Hohenferner glacier forefield) and annual precipitation (third epoch: 2006-2019) in the Horlachtal
(Grastalferner glacier forefield) are statistically significant, although the latter both cover only 13 and 14 years. In the
Horlachtal, the first two epochs show a decreasing trend in precipitation, while the third epoch shows an increase in
precipitation, which is significantly more pronounced in summer than in winter. In the Kaunertal, winter precipitation shows
a slight increase in the first epoch and a stronger decrease in summer precipitation. The second epoch shows a slight increase
in summer and a slight decrease in winter. The third epoch also shows a strong decrease in summer and winter precipitation.





In the Martelltal, on the other hand, winter precipitation decreases significantly and summer precipitation increases
significantly, especially in the second epoch, although epochs of different lengths are analysed.

**Figure 10: Mean annual, mean summer and mean winter precipitation of the respective glacier forefields.**



In the following, the changes of different precipitation classes (as well as with a different temporal resolution) between the
individual epochs are analysed. The calculated changes are based on Appendix B and C. In these tables, the number of all
precipitation events of the corresponding epochs is shown and divided into corresponding precipitation classes. Using the
number of years per epoch, this results in an mean annual number of events per class. The calculated changes result from the
comparison of the mean occurrence of the precipitation classes of the previous epoch. Both precipitation events with a
resolution of one hour (Figure 11/Appendix B) and daily precipitation totals (Figure 12/Appendix C) were analysed. The
highest temporal resolution (1 hour) shows that the classes >0 to 4 and 4 to 8 are subject to the highest variations (Figure 11);
for example, precipitation events of the class >0 to 4 occur 7.63 times less in the glacier forefield of the Grastalferner in the
second epoch compared to the first. In general, it can be seen that the higher precipitation classes tend to decrease, albeit very
slightly, but there are still changes with both an increase and a decrease in the different precipitation classes. The daily
precipitation totals also show a high variation, with both a decrease and an increase over the different epochs (Figure 12). In
general, there are also very slight changes. Nevertheless, the decrease in the higher three classes predominates when comparing
the third with the second epoch.





**Figure 11: Change in precipitation classes between the different epochs with a one-hour resolution (extreme precipitation events).**





**Figure 12: Change in precipitation classes between the different epochs with a 24-hour resolution (daily precipitation totals).**

 

The analysis of the mean annual rainfall and snowfall events shows that there is a consistent increase in rainfall and at the
same time a consistent decrease in snowfall (except for the Weißseeferner glacier forefield) (Figure 13).

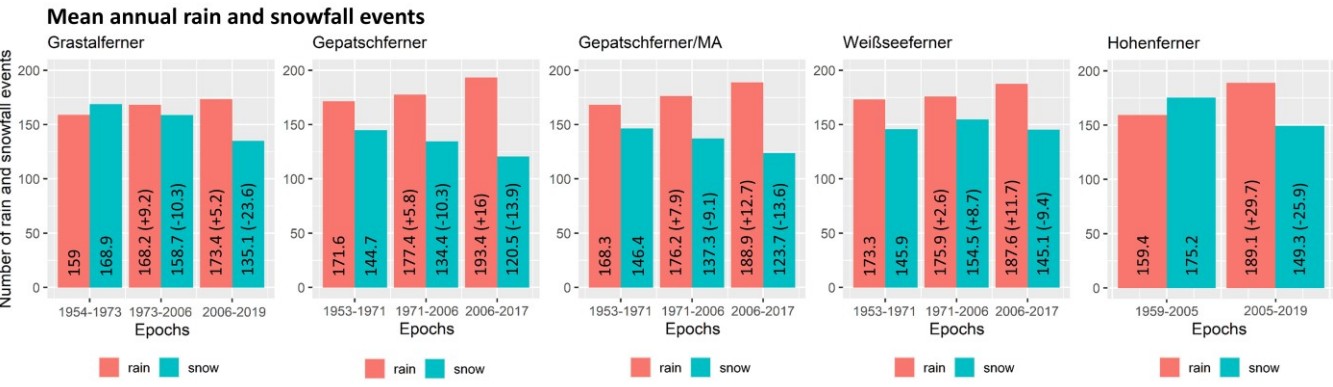


**Figure 13: Mean annual rain and snowfall events with the corresponding changes between epochs.**
**5 Discussion**
**5.1 Assessment of the SCA approach**
Using the relationship between accumulated SY from DoDs and SCA/CA (log-log model) of different AOIs and different
epochs, we show a long-term monitoring of several geomorphologically active sections of LIA lateral moraines. This is a clear
difference to previous studies that used the space-for-time-substitution (SFTS) approach in proglacial areas, in which studies
used recent morphometrical or morphodynamical differences between sites located along a gradient of deglaciation age to
infer long-term changes in morphodynamics (Ballantyne and Benn, 1994; Curry, 1999; Curry et al., 2006). Means that long-
term studies with quantitative data are rare (Schiefer and Gilbert, 2007; Betz et al., 2019; Altmann et al., 2020; Betz-Nutz,
2021; Betz-Nutz et al., 2022). The approach shown here provides reliable results and requires only a few input data (Neugirg
et al., 2015a; 2015b; 2016; Dusik, 2019; Dusik et al., 2019). The results mainly show a decrease in SY as well as a decrease
in the slope of the regression lines (suggesting less SY per square metre) over the different epochs, indicating a decrease in
geomorphic activity on these AOIs. In some AOIs, we observe contrasting changes: There is an increase in SY and an increase
in the slope of the regression line at HG1 and almost no change in SY on the Y-axis but an increase in the slope of the regression
line at MH1, which can be described as an increase (HG1) and a constant geomorphic activity (MH1). Moreover, in the earlier
epochs, a clearly higher variability of SY (Slope of the regression line clearly higher 1) was observed on the respective AOIs,
which is no longer reached in the later ones. Thus, it is possible to describe two different types of change in SY (size of SY
between epochs and variability of SY within an epoch, which can also be compared with the other epochs). Slopes of the
regression line below 1 could occur when spots appear within the area that are no longer active, which could be an indication
of stabilisation, which occurs mainly in the second and third epoch.





The p-values of the coefficients are mostly below the alpha level of 0.05, so it is assumed that the relationships between SCA
and SY are statistically significant in almost all cases (~92%) (Figure 6). To determine the proportion of the variance of the
dependent variable that can be explained by the independent variable, the R-squares (R² or the coefficient of determination) of
all regression lines were analysed (Figure 6, Appendix A). The relationship between SCA and SY shows varying correlations
within the AOI and the epochs. The median R² values range from 0.59 to 0.91 in the first epoch, from 0.37 to 0.93 in the second
epoch and from 0.3 to 0.94 in the third epoch (Figure 6, Appendix A). The number of channels modelled differed between the
epochs on the same AOIs due to the different quality of the DEMs and the slightly different size of these. As in Heckmann
and Vericat (2018), the accumulation of DoD values resulted in very small positive values at some AOIs. Such errors are due
to the quality of the DoD, different bulk densities of eroded vs. deposited materials, and the inability of the flow routing
algorithm to fully reproduce sediment transfer in reality especially when flow directions changed within one epoch. Where
positive accDoD values occurred, they were small and manually corrected to the zero.
Nevertheless, the D8 algorithm simplifies complex sediment transport processes such as fluvial activity, landslides and debris
flows, which have different frequencies, magnitudes and forms of erosion and accumulation. As the individual epochs cover
several years, no reference can be made in this study to individual processes that can be attributed to extreme precipitation
events or to seasonal differences. Therefore, we compare different epochs based on mean annual SY, which includes all
geomorphological processes. Accordingly, the aim was not to model individual erosion processes but to compute SY of each
cell. The individual AOIs have a slightly different area within the different epochs, which is mainly due to headcut retreat
(Heckmann and Vericat, 2018; Betz-Nutz et al., 2022). The lateral boundaries also changed slightly due to the quality of the
DEMs and geomorphological slope processes, while the lower boundary did not change.
By processing historical aerial photographs into DEMs (by SfM-MVS), the temporal aspect of SCA studies could be quickly
and cost-effectively extended to several decades (up to the 1950s), which previously spanned only a few months or several
years (Neugirg et al., 2015a; 2015b; 2016; Dusik, 2019; Dusik et al., 2019). However, as Schiefer and Gilbert (2007) have
already shown, the shorter the time intervals and the lower the quality of the aerial images, the more difficult it becomes to
detect surface changes, so in the process of this study several series of aerial images had to be sorted out that were actually
available due to a poor data quality. Furthermore, it should be noted that the accuracy and precision of the historical DEMs
strongly depends on the respective generation, e.g. whether they were generated with or without a calibration certificate (as
was the case, for example, with the 1959 aerial photo series in the Horlachtal/glacier forefield Hohenferner), which ultimately
influences the SCA results and the calculated erosion rates (Stark et al., 2022).
**5.2 Geomorphic activity**
The geomorphic activity is directly related to the characteristics of the AOIs. The AOIs with the highest mean annual sediment
output (>100m³/a) (such as KG1, KG2, KG4, KG5, MH2, KM2, KW2) show strong gully formation and are overall
characterised by larger areas, longer max. slope lengths and higher mean and max. slope gradients (Table 1). In contrast, the
AOIs with lower mean annual sediment output (<100m³/a) (such as KW1, KG3, HG1, KM1, MH1) show less gully incision.



These AOIs tend to be characterised by smaller areas, smaller max. slope lengths and smaller mean slope and max. slope
gradients (Table 1). The strong influence of slope length and slope gradient on morphodynamics is also shown by previous
studies (Ballantyne and Benn, 1994; Curry, 1999; Curry et al., 2006; Betz-Nutz et al., 2022). KG3 also appears to be somehow
stabilized by bedrock in the lower part of the slope, which could mitigate the erosion of this AOI. Elevation and aspect,
however, do not seem to have an influence on geomorphic activity, which is also shown in the study by Curry et al. (2006).
Since only bare and sparsely vegetated areas were investigated, no findings on the influence of vegetation on morphodynamics
can be made in this study. Solifluction processes could also not be observed, probably due to the composition of the moraine
material. Presumably, the morphodynamics are still so high that the vegetation does not yet have the opportunity to develop
accordingly. In general, we assume that debris flows are the most common process, as described for example by Ballantyne
(2002a) and Curry et al. (2006). Thus, material stored in the gullies is transported downslope by debris flows mainly rain or
snow events in the spring or heavy rainfall events during rainstorms in the summer months (Ballantyne and Benn, 1994;
Ballantyne, 2002b; Curry et al., 2006; Dusik et al., 2019).
However, the high mean annual sediment yield and corresponding erosion rate in the first (3471 m³/a, 465 mm/a) and second
(2922 m³/a, 245 mm/a) epochs of AOI KG1 (Figure 7) can probably also be attributed to individual landslides and deep-seated
slope failures in some cases linked with melting dead ice bodies, as these processes are more likely to occur after deglaciation,
and are characterised by high magnitude and low frequency, which has also been shown by Blair (1994), Hugenholtz et al.
(2008) and Cody et al. (2020). On the less incised slopes (e.g. MH1), small-scale processes such as fluvial erosion or snow
drifts probably occur (Betz-Nutz, 2021), which ultimately show no clear trend in the increase or decrease of morphodynamics,
but can be described as a constant geomorphic activity. In Betz-Nutz et al. (2022) and in this study, six similar lateral moraine
sections (although other exactly defined AOIs) were investigated. The test sites KG1, KG2, KW1, KW2 and MH1 (in Betz-
Nutz et al. (2022): GPF1, GPF2, WSF1, WSF2 and HF1) showed similar erosion rates and the same log-term trends. In the
case of AOI MH2, different trends were determined (stagnation in Betz-Nutz et al. (2022) and a decrease in this study), which
can be attributed to the differently defined AOI and the slightly different study period.
In the sense of a process-response system, it is noticeable that the first-mentioned group of AOIs with the higher erosion rates
(except KG1) had considerable influence from melting dead ice in the lower slope area at least until 2006 (KG2, KM2, KW2,
MH2) or the glacier was still present at the bottom of the slope (KG4, KG5), which could be identified by the interpretation of
the DoDs. Melting of the dead ice can lead to destabilisation of the slope, which can enhance erosion processes of the upper
slope areas, as the support is no longer present, the sediment becomes saturated and there can be an increase in the slope
gradient due to the subsidence of the lower part of the slope (Altmann et al., 2020; Betz-Nutz et al., 2022). However, the
highest slope gradients are also present here, which also plays a major role. In addition, AOI HG1, where erosion is increasing,
shows an undercutting of the slope by the adjacent stream, which leads to a destabilization or lowering of the erosion base and
a typical formation of a debris cone and alluvial fan with a successive reduction of the slope gradient is missing (Figure 14).
It can be assumed that individual strong rainfall events in the second and third epochs in combination with changing flow
pathes due to the retreat of the Grastalferner acted here as an impulse and affected both the AOI itself and the adjacent stream.



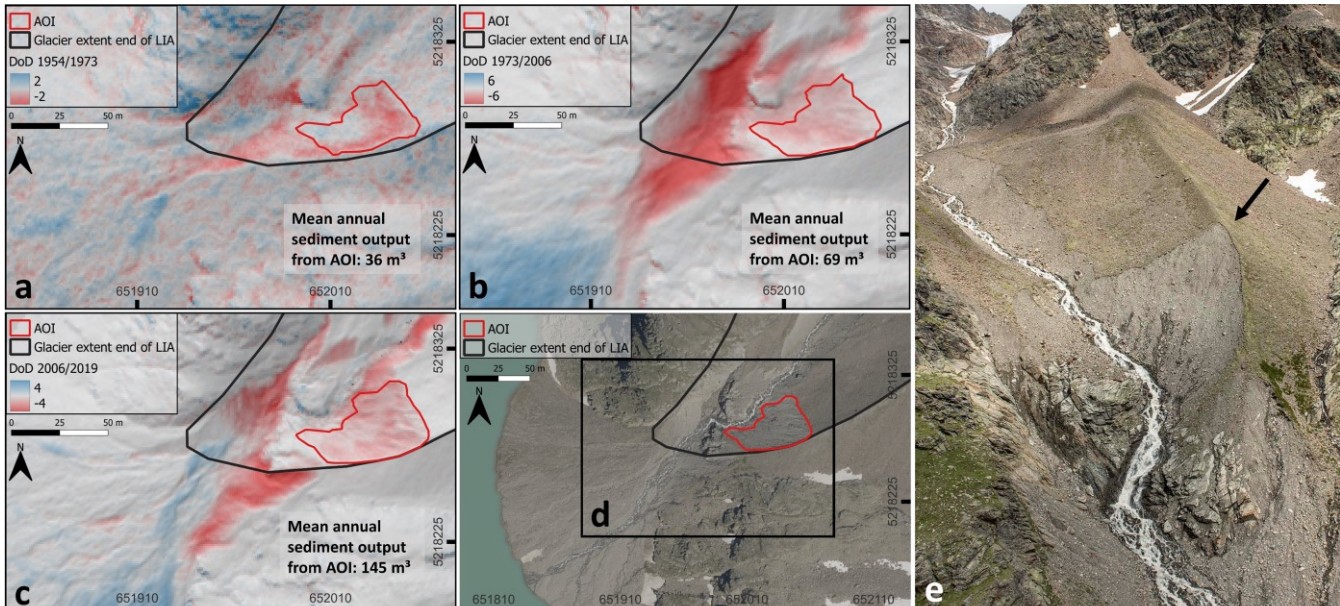

**Figure 14: Overview of the DoDs of the corresponding epochs of AOI HG1: (a) DoD 1954-1973, (b) DoD 1973-2006, (c) DoD 2006-2019, (d) Orthofoto 2020 (provided by the Province of Tyrol) and (e) photo of the AOI from 2019 by Anton Brandl.**

## 5.3 Paraglacial landscape adjustment

### 5.3.1 The "Sediment activity concept"

The finding of mainly decreasing geomorphic activity of LIA lateral moraines in this study is largely consistent with previous model-based studies describing the paraglacial landscape adjustment with a decrease in geomorphic activity in proglacial areas over time, such as the theoretical model "paraglacial concept" of Church and Ryder (1972) or the "sediment exhaustion model" of Ballantyne (2002a, 2002b). The geomorphic activity of gully systems is given as a few decades to centuries (Ballantyne, 2002a; 2002b). Furthermore, it is stated that there is a high temporal and spatial variability in this development. The model provides an appropriate approximation (Ballantyne, 2002a; 2002b). Within the paraglacial adjustment process, different geomorphological processes result in different durations of occurrence. Furthermore, different land systems react at different rates and on different spatial scales. Thus, external perturbations can occur, leading to secondary peaks and time delays (Ballantyne, 2002a; 2002b). While ten out of twelve AOIs fit the model descriptions, two test plots show opposite morphodynamics, which can be described as a delay of the paraglacial adjustment process or that response systems can run counter to such an adaption.

To estimate the changing morphodynamics, we therefore propose the following simplified description of the landscape evolution using the "Sediment activity concept" based on the results of this study. Due to the study design, this concept is only valid on geomorphologically active areas (in this case the upper lateral moraine section) on LIA lateral moraines and until about 170 years after the end of LIA (Table 6, Figure 15). The concept distinguishes between an earlier and a later phase. The



earlier phase (mainly 1950s to 1970s) is characterised by a wide range between areas with high and low variability of SY
within the area as well as high and low mean annual sediment yield (erosion rate/volume). In contrast, the later stadium (mainly
1970s to 2000s and 2000s to the end of the survey 2017/2019) shows a decrease in this range. Although the decrease of
morphodynamics predominates, there are also increases in morphodynamics. The time of ice release was not integrated here,
so that the time periods refer to the actual time. In addition, we give two examples in Figure 15. While example A shows high
variability (polynomial behaviour) within the area and high SY (erosion rates/sediment output), example B shows low
variability (constant/linear behaviour) and low SY (erosion rates/sediment output). Ultimately, the relationship between SY
and the size of the catchment has changed so that erosion within the area is more constant today.

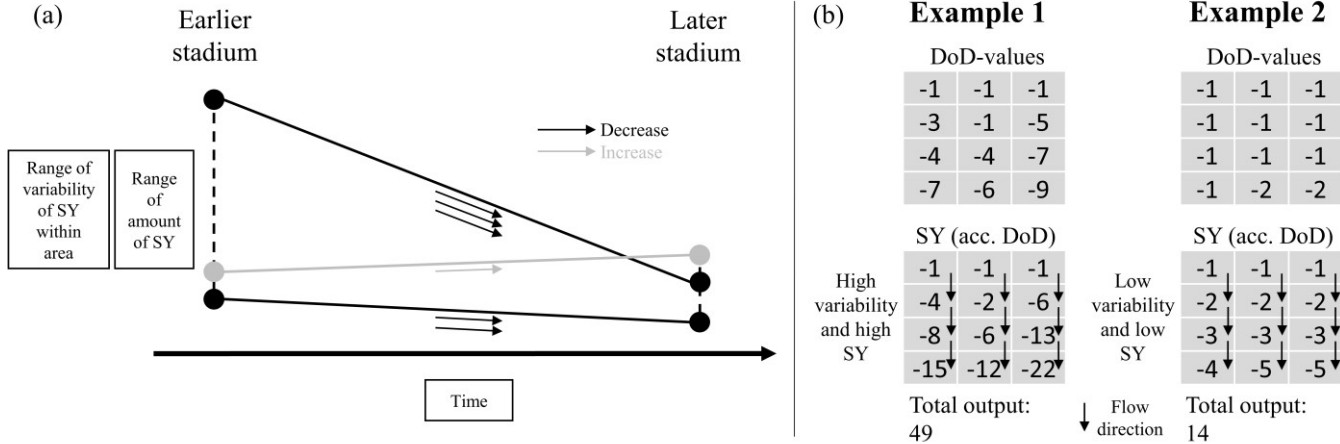


**Figure 15: The "Sediment activity concept". Description and illustration of the change in sediment activity over time (a) and 2**
**corresponding examples (b).**
**Table 6: Tabular summary of the simplified conceptual model.**

| | Earlier stadium | Later stadium |
|---|---|---|
| SY within area | highly variable up to constant | in the range of constant |
| Amount of SY | wide range | lower range |
| Over time | | mostly decreasing |


Introducing the "Sediment activity concept" of this study, we present a different description of the paraglacial adjustment
process which is based on the actual SY. However, the concept is only valid until about 170 years after the end of LIA and on
the AOIs of this study. Nevertheless, we assume that the concept can also prove its validity in further proglacial active areas
and partly over an even longer period of time.



### 5.3.2 Erosion rates

The comparison of long-term erosion rates of gully systems in proglacial areas of different studies shows the high variability of this adjustment; as these studies were carried out with different methods, on different time scales and in different glacier forefields, comparing their results is difficult (Table 7). The methodologies for determining long-term average erosion rates are based on gully volume estimates (Ballantyne and Benn, 1994; Curry, 1999; Curry et al., 2006) and sediment volume calculations due to surface changes using DoDs, as shown by Betz-Nutz et al. (2022) and this study. In glacier forefields in western Norway, this amounts e.g. to minimum estimates of 50-100 mm/year (max. estimates of min. 200 mm/year) (Ballantyne and Benn, 1994) and a minimum of 5.5-169 mm/year (in different glacier forefields) (Curry, 1999). A further study in the Swiss Alps shows erosion rates of min. 49-151 (in different glacier forefields) (Curry et al., 2006). The work of Betz-Nutz et al. (2022) and this study show erosion rates over several decades and distinguish between different epochs, which makes it possible to show differences between them. Both studies show that the mean erosion rates in the individual epochs decrease (Table 7), although in individual cases there is also a constant and an increase in erosion rates over time. Although there has been a clear decrease in geomorphic activity, stabilisation of the AOIs is not yet apparent, which means that the paraglacial adjustment is still ongoing. Within this study, we observed that the AOIs still show a high geomorphic activity even after they have been deglaciated for 76-159 years. A stabilisation of the gully systems as shown by Curry (2006) cannot be observed. Other studies such as Lane et al. (2017), Dusik (2019), Altmann et al. (2020), Betz-Nutz et al. (2022) also show the still ongoing paraglacial adjustment processes.

**Table 7: Studies on long-term erosion rates (several decades) of gully systems on LIA lateral moraines in different glacier forefields.**

| Study | Erosion rate (mm/year) | Timescale (year) | Time since ice exposure (year) | Location of the study area |
|---|---|---|---|---|
| Ballantyne and Benn (1994) | Min. of 50-100, max. min. of 200 | 48 | 48 | Norway, Fåbergstølsbreen |
| Curry (1999) | Min. of 5.5-8.8, 38-169 and 19-169 | 76, 53, 43 | 76, 53, 43 | Norway, Fåbergstølsbreen, Lodalsbreen and Heillstugubreen |
| Curry et al. (2006) | Min. of 86-151 and 49-103 | 55, 79 | 55, 79 | Switzerland, Glacier du Mont Miné and Feegletscher |
| Betz-Nutz (2021; 2022) | Epoch I: 2-429, epoch II: 1-186, epoch III: 3-110 | Epoch I: Mainly ~1950s to ~1970s, Epoch II: ~1970s to ~2000s and Epoch III: ~2000s to 2018/2019 | 59-154 | Austria (Tyrol), Germany (Bavaria) and Italy (South |



| This study | Epoch I: 19-465, epoch II (HT and KT): 13-245, epoch II (MT) and epoch III (HT and KT): 8-88 | Epoch I: 17-19 (in HT and KT) and 45/46 (in MT), epoch II: 33-36 (in HT and KT) and 14/15 (in MT) and epoch III: 11-13 (in HT and KT) | 76-159 | Tyrol), ten different glacier forefields Austria (Tyrol) and Italy (South Tyrol), five different glacier forefields |

*HT = Horlachtal, KT = Kaunertal and MT = Martelltal.

## 5.4 Meteorological drivers

The decrease of the mean annual number of ice days and the increase in the number of warm spells over the different epochs and the associated potential increase in snowmelt on the slopes could also lead to an increase in morphodynamics, as these processes represent important preparatory steps for erosion processes in spring (Haas, 2008), such as increased saturation of the slope du to snow melt, loosening of the upper sediment layers or the delivery of material by snow slides or small wet avalanches that is then available for debris flows in the summer months (Dusik et al., 2019). Klein et al. (2016), for example, also show an increase in the frequency and intensity of snowmelt in the Swiss Alps. Mean annual precipitation decreases slightly across epochs, but is not statistically significant (except for winter precipitation for the entire study period (1959 to 2019) and summer precipitation in the second epoch from 2005 to 2019 in the Martelltal). Other studies also show that the decrease in precipitation in the Alps is low (Brugnara et al., 2012) and that there is no clear trend in precipitation (Hock et al., 2019) or that it is mainly subject to regional influences and decadal variations (Mankin and Diffenbaugh, 2015). Extreme precipitation events (1h resolution) and daily precipitation totals also show only minor changes. Differentiation of precipitation, on the other hand, shows a clear increase in rainfall and a decrease in snowfall, which is also shown by Serquet et al. (2011), Beniston et al. (2018) and Hock et al. (2019) who found that the rainfall on snow events in spring as preperatory factor for the erosion processes in the summer months increase. The simulated meteorological data generally show lower temperatures and larger precipitation amounts, when compared to three automatic weather stations operated by TIWAG (Tyrolean Hydropower AG, Innsbruck, Austria). These stations are located in the vicinity of our AOIs. The simulated mean annual temperatures extracted at the location of the weather stations Horlachalm (1987-2015) (approx. 6.5 km linear distance to the AOI in the Grastalferner glacier forfield) and Weißseeferner (2007-2015) (approx. 500 m linear distance to the AOIs in the Weißseeferner glacier forfield), covering the same time period indicate a difference of -1.05°C and of -0.87°C, respectively, after accounting for differences in elevation. However, at Gepatschalm (2010-2015) weather station (approx. 2.5 km linear distance to the AOIs in the Gepatschferner glacier forfield), the difference between the simulated and observed mean annual temperatures is 0.13°C, indicating that the magnitude of the discrepancies depends on the station data used for the comparison. The simulated precipitation, however, is generally larger with mean annual precipitation sums of 1531, 1655, and 1820 mm at the location of Horlachalm (1990-2015), Gepatchalm (2010-2015), and Weißseeferner (2007-2015), respectively, while the





weather stations recorded values of 803, 1086, and 924 mm, indicating large discrepancies especially when compared to
Horlachalm and Weißseeferner weather stations. The datasets from which the temperature and precipitation were extracted are
both based on coarsely resolved data, which makes a comparison with measurement data in the field difficult, although the
corresponding trends are well usable for this study. The large difference between simulated and recorded precipitation is mainly
due to winter precipitation (Figure 11) when the weather stations are not always able to record total snowfall accurately;
additionally, fog precipitation or precipitation in combination with stronger winds are not recorded correctly.
The weather and climate study periods are based on the predefined study epochs given by the availability and quality of
orthophotos, and not on the usual climate periods. This results in large differences in the length of the different epochs, which
must be taken into account.
There are several sources of uncertainty in the simulated data, amongst them the dynamic initial and boundary conditions, as
the forcing data have their own sources of uncertainties. Furthermore, the choice of the reanalysis data used for forcing the
model has an influence on the final results. Additionally, for such long simulations, an updated sea surface temperature (SST)
is recommended. Since there are no SSTs available for the 20CRv3, we have generated SST fields from the skin temperature.
Other sources of uncertainty are the static boundary conditions like the fixed land use categories and topography, as well as
model simplifications and choices in the parameterization of the physics and dynamics. In our simulations, we have used
spectral nudging in order to keep the model from large deviations from the forcing data. Short test runs indicate that the use of
spectral nudging improves the simulated data, especially with respect to precipitation. However, the strength of nudging also
has an influence on the final resuls. Since the purpose of this study is not to test how strongly to nudge, we have used the
default values in WRF.
**6 Conclusion**
Using DoDs based on SfM photogrammetric and LiDAR data DEMs, we show with two different approaches, the long-term
(1953-2019) change in the morphodynamics of several active gully systems on LIA lateral moraines in the Tyrolean and South
Tyrolean Alps, Austria and Italy. First, the change in the range of variability of SY within the area (using regression lines with
accSY and SCA/CA) and second, the change in the amount of SY (calculation of erosion rates/volume of sediment output)
between the different epochs could be shown.
Finally, the first epoch shows a clearly higher range of variability of SY within the AOI than the later epochs. This means that
the spatial pattern of erosion has become more uniform within the areas. In addition, the total sediment yield, the mean annual
sediment yield and the mean annual specific sediment yield (erosion rate) were calculated for each AOI and epoch was
calculated. Over the epochs, there is a decreasing trend of geomorphological activity in 10 out of 12 AOIs, while 2 AOIs show
an opposite trend, where morphodynamics increase or remain at the same level. Overall, we confirm the general trend of
decreasing morphodynamics over time (10 AOIs) of several previous studies, although we could also show that the geomorphic
activity of one AOI is on the same level and one is increasing. Finally, the results led to the proposal of a simplified conceptual





model "The sediment activity concept", describing the paraglacial adjustment process by summarising the findings on the
long-term morphodynamics of the upper parts (gully heads) of lateral moraines from this study.
Despite the general decline in morphodynamics, the AOIs show no stabilisation, leading us to the conclusion that the
paraglacial landscape adjustment is still in progress (even on areas that have been ice-free for at least 159 years). It seems that
the vegetation has not yet had the opportunity to develop due to the high morphodynamics. In general, debris flows are probably
the most common processes, although it is difficult to separate the different processes, but very high SY (mainly in the first
epoch) also indicate landslides and slope failures. AOI morphodynamic is also related to the characteristics, i.e. AOIs that are
larger, have longer max. lengths and higher mean slope gradients (as well as max. slope gradients) have clearly higher
geomorphic activity and form more deeply incised gullies. In the sense of a process-response system, it can be stated that the
melting of dead ice in the lower slope area, which in some cases lasts for decades, leads to high morphodynamics of the upper
slope area. Furthermore, it is assumed that the lowering of the erosion base by adjacent streams leads to a delay of the
paraglacial landscape adjustment, as the formation of an accumulation area is disrupted.
In addition to the system-internal influences on morphodynamics, we assume an additional influence of changing weather and
climate factors on the corresponding erosion processes with an increase (mainly in the last, i.e. most recent epoch from the
mid-2000s to 2017/2019), since the statistically significant warming of the last decades has led to a reduction of the mean
annual ice days, to an increase in warm air inflows and, when distinguishing between rainfall and snowfall, to an increase in
rainfall. We do not see any clear influence in the changing precipitation, although it can be assumed that the same precipitation
intensities led to higher erosion in the first epoch than in the second or third. Nevertheless, the system-internal dynamics and
the general paraglacial adaptation process seem to have the greatest impact on the changing morphodynamics. Future work
should apply the approach used here to more areas and, if possible, with a higher temporal resolution to improve the process
understanding of erosion on lateral moraines.





## Appendix A

Boxplots of the model parameters Intercept, Slope and R² of all regression lines (see Figure 6).





**Appendix B**
**Calculation of the individual extreme events by continuous ongoing 4 mm classes (with one-hour resolution).**

| | Precipitation interval (mm) | No. events/total | No. events/year | No. events/total | No. events/year | No. events/total | No. events/year | Change from previous epoch | |
|---|---|---|---|---|---|---|---|---|---|
| Grastalferner (Horlachtal) | | 1954-1973 (epoch I) | | 1973-2006 (epoch II) | | 2006-2019 (epoch III) | | I to II | II to III |
| | >0 to 4 | 6084 | 320.21 | 10315 | 312.58 | 4056 | 312.00 | -7.63 | -0.58 |
| | 4 to 8 | 368 | 19.37 | 615 | 18.64 | 194 | 14.92 | -0.73 | -3.71 |
| | 8 to 12 | 78 | 4.11 | 124 | 3.76 | 49 | 3.77 | -0.35 | 0.01 |
| | 12 to 16 | 17 | 0.89 | 33 | 1.00 | 12 | 0.92 | 0.11 | -0.08 |
| | 16 to 20 | 7 | 0.37 | 13 | 0.39 | 4 | 0.31 | 0.03 | -0.09 |
| | 20 to 24 | 2 | 0.11 | 8 | 0.24 | 2 | 0.15 | 0.14 | -0.09 |
| | 24 to 28 | 1 | 0.05 | 2 | 0.06 | 1 | 0.08 | 0.01 | 0.02 |
| | >28 | 1 | 0.05 | 1 | 0.03 | 0 | 0.00 | -0.02 | -0.03 |
| Gepatschferner (Kaunertal) | | 1953-1971 (epoch I) | | 1971-2006 (epoch II) | | 2006-2017 (epoch III) | | I to II | II to III |
| | >0 to 4 | 5498 | 305.44 | 10278 | 293.66 | 3447 | 313.36 | -11.79 | 19.71 |
| | 4 to 8 | 390 | 21.67 | 700 | 20.00 | 254 | 23.09 | -1.67 | 3.09 |
| | 8 to 12 | 87 | 4.83 | 178 | 5.09 | 47 | 4.27 | 0.25 | -0.81 |
| | 12 to 16 | 20 | 1.11 | 44 | 1.26 | 14 | 1.27 | 0.15 | 0.02 |
| | 16 to 20 | 9 | 0.50 | 15 | 0.43 | 4 | 0.36 | -0.07 | -0.06 |
| | 20 to 24 | 3 | 0.17 | 9 | 0.26 | 1 | 0.09 | 0.09 | -0.17 |
| | 24 to 28 | 1 | 0.06 | 1 | 0.03 | 0 | 0.00 | -0.03 | -0.03 |
| | >28 | 1 | 0.06 | 0 | 0.00 | 0 | 0.00 | -0.06 | 0.00 |
| Gepatschferner/ Müchner/ Abfahrt (Kaunertal) | | 1953-1971 (epoch I) | | 1971-2006 (epoch II) | | 2006-2017 (epoch III) | | I to II | II to III |
| | >0 to 4 | 5452 | 302.89 | 10316 | 294.74 | 3428 | 311.64 | -8.15 | 16.89 |
| | 4 to 8 | 402 | 22.33 | 750 | 21.43 | 256 | 23.27 | -0.90 | 1.84 |
| | 8 to 12 | 96 | 5.33 | 165 | 4.71 | 51 | 4.64 | -0.62 | -0.08 |
| | 12 to 16 | 23 | 1.28 | 41 | 1.17 | 10 | 0.91 | -0.11 | -0.26 |
| | 16 to 20 | 4 | 0.22 | 8 | 0.23 | 4 | 0.36 | 0.01 | 0.14 |
| | 20 to 24 | 1 | 0.06 | 6 | 0.17 | 2 | 0.18 | 0.12 | 0.01 |
| | 24 to 28 | 2 | 0.11 | 1 | 0.03 | 0 | 0.00 | -0.08 | -0.03 |
| Weißseeferner (Kaunertal) | | 1953-1971 (epoch I) | | 1971-2006 (epoch II) | | 2006-2017 (epoch III) | | I to II | II to III |
| | >0 to 4 | 5690 | 316.11 | 10580 | 302.29 | 3506 | 318.73 | -13.83 | 16.44 |
| | 4 to 8 | 408 | 22.67 | 745 | 21.29 | 255 | 23.18 | -1.38 | 1.90 |
| | 8 to 12 | 96 | 5.33 | 166 | 4.74 | 56 | 5.09 | -0.59 | 0.35 |
| | 12 to 16 | 23 | 1.28 | 44 | 1.26 | 7 | 0.64 | -0.02 | -0.62 |
| | 16 to 20 | 9 | 0.50 | 14 | 0.40 | 5 | 0.45 | -0.10 | 0.05 |
| | 20 to 24 | 2 | 0.11 | 8 | 0.23 | 2 | 0.18 | 0.12 | -0.05 |
| Hohenferner (Martelltal) | | 1959-2005 (epoch I) | | | | 2005-2019 (epoch II) | | I to II | |
| | >0 to 4 | 14531 | 315.89 | | | 4757 | 339.79 | 23.89 | |
| | 4 to 8 | 883 | 19.20 | | | 240 | 17.14 | -2.05 | |
| | 8 to 12 | 241 | 5.24 | | | 59 | 4.21 | -1.02 | |
| | 12 to 16 | 57 | 1.24 | | | 17 | 1.21 | -0.02 | |
| | 16 to 20 | 8 | | | | 2 | 0.14 | -0.03 | |
| | 20 to 24 | 3 | 0.07 | | | 1 | 0.07 | 0.01 | |
| | 24 to 28 | 3 | 0.07 | | | 0 | 0.00 | -0.07 | |




**Appendix C**
**Calculation of the daily precipitation totals by continuous ongoing 4 mm classes of the different epochs (24-hour resolution).**

| Glacier forefield | Precipitation interval (mm) | No. events/total | No. events/year | No. events/total | No. events/year | No. events/total | No. events/year | Change from previous epoch | |
|---|---|---|---|---|---|---|---|---|---|
| Grastalferner | | 1954-1973 (epoch I) | | 1973-2006 (epoch II) | | 2006-2019 (epoch III) | | I to II | II to III |
| (Horlachtal) | 0 to 10 | 410 | 21.58 | 732 | 22.18 | 271 | 20.85 | 0.60 | -1.34 |
| | 10 to 20 | 181 | 9.53 | 341 | 10.33 | 139 | 10.69 | 0.81 | 0.36 |
| | 20 to 30 | 113 | 5.95 | 191 | 5.79 | 70 | 5.38 | -0.16 | -0.40 |
| | 30 to 40 | 47 | 2.47 | 101 | 3.06 | 41 | 3.15 | 0.59 | 0.09 |
| | 40 to 50 | 26 | 1.37 | 35 | 1.06 | 9 | 0.69 | -0.31 | -0.37 |
| | 50 to 60 | 13 | 0.68 | 14 | 0.42 | 7 | 0.54 | -0.26 | 0.11 |
| | 60 to 70 | 0 | 0.00 | 6 | 0.18 | 1 | 0.08 | 0.18 | -0.10 |
| | >70 | 2 | 0.11 | 0 | 0.00 | 1 | 0.08 | -0.11 | 0.08 |
| Gepatschferner | | 1953-1971 (epoch I) | | 1971-2006 (epoch II) | | 2006-2017 (epoch III) | | I to II | II to III |
| (Kaunertal) | 0 to 10 | 335 | 18.61 | 706 | 20.17 | 218 | 19.82 | 1.56 | -0.35 |
| | 10 to 20 | 174 | 9.67 | 325 | 9.29 | 108 | 9.82 | -0.38 | 0.53 |
| | 20 to 30 | 105 | 5.83 | 224 | 6.40 | 67 | 6.09 | 0.57 | -0.31 |
| | 30 to 40 | 73 | 4.06 | 107 | 3.06 | 36 | 3.27 | -1.00 | 0.22 |
| | 40 to 50 | 22 | 1.22 | 50 | 1.43 | 19 | 1.73 | 0.21 | 0.30 |
| | 50 to 60 | 11 | 0.61 | 19 | 0.54 | 4 | 0.36 | -0.07 | -0.18 |
| | 60 to 70 | 5 | 0.28 | 11 | 0.31 | 1 | 0.09 | 0.04 | -0.22 |
| | 70 to 80 | 2 | 0.11 | 3 | 0.09 | 0 | 0.00 | -0.03 | -0.09 |
| | 80 to 90 | 0 | 0.00 | 2 | 0.06 | 0 | 0.00 | 0.06 | -0.06 |
| | 90 to 100 | 0 | 0.00 | 5 | 0.14 | 2 | 0.18 | 0.14 | 0.04 |
| | >100 | 4 | 0.22 | 5 | 0.14 | 0 | 0.00 | -0.08 | -0.14 |
| Gepatschferner/ | | 1953-1971 (epoch I) | | 1971-2006 (epoch II) | | 2006-2017 (epoch III) | | I to II | II to III |
| Müchner | 0 to 10 | 304 | 16.89 | 618 | 17.66 | 202 | 18.36 | 0.77 | 0.71 |
| Abfahrt | 10 to 20 | 146 | 8.11 | 301 | 8.60 | 96 | 8.73 | 0.49 | 0.13 |
| (Kaunertal) | 20 to 30 | 101 | 5.61 | 197 | 5.63 | 64 | 5.82 | 0.02 | 0.19 |
| | 30 to 40 | 69 | 3.83 | 115 | 3.29 | 36 | 3.27 | -0.55 | -0.01 |
| | 40 to 50 | 35 | 1.94 | 50 | 1.43 | 21 | 1.91 | -0.52 | 0.48 |
| | 50 to 60 | 12 | 0.67 | 26 | 0.74 | 6 | 0.55 | 0.08 | -0.20 |
| | 60 to 70 | 5 | 0.28 | 12 | 0.34 | 1 | 0.09 | 0.07 | -0.25 |
| | 70 to 80 | 1 | 0.06 | 5 | 0.14 | 0 | 0.00 | 0.09 | -0.14 |
| | >80 | 4 | 0.22 | 10 | 0.29 | 2 | 0.18 | 0.06 | -0.10 |
| Weißseeferner | | 1953-1971 (epoch I) | | 1971-2006 (epoch II) | | 2006-2017 (epoch III) | | I to II | II to III |
| (Kaunertal) | 0 to 10 | 302 | 16.78 | 623 | 17.80 | 212 | 19.27 | 1.02 | 1.47 |
| | 10 to 20 | 146 | 8.11 | 296 | 8.46 | 104 | 9.45 | 0.35 | 1.00 |
| | 20 to 30 | 99 | 5.50 | 203 | 5.80 | 60 | 5.45 | 0.30 | -0.35 |
| | 30 to 40 | 75 | 4.17 | 104 | 2.97 | 34 | 3.09 | -1.20 | 0.12 |
| | 40 to 50 | 32 | 1.78 | 46 | 1.31 | 19 | 1.73 | -0.46 | 0.41 |
| | 50 to 60 | 12 | 0.67 | 28 | 0.80 | 6 | 0.55 | 0.13 | -0.25 |
| | 60 to 70 | 3 | 0.17 | 14 | 0.40 | 1 | 0.09 | 0.23 | -0.31 |
| | 70 to 80 | 1 | 0.06 | 3 | 0.09 | 1 | 0.09 | 0.03 | 0.01 |
| | 80 to 90 | 1 | 0.06 | 4 | 0.11 | 1 | 0.09 | 0.06 | -0.02 |
| | >90 | 3 | 0.17 | 6 | 0.17 | 0 | 0.00 | 0.00 | -0.17 |
| Hohenferner | | 1959-2005 (epoch I) | | | | 2005-2019 (epoch II) | | I to II | |
| (Martelltal) | 0 to 10 | 1169 | 25.41 | | | 387 | 27.64 | 2.23 | |
| | 10 to 20 | 403 | 8.76 | | | 127 | 9.07 | 0.31 | |
| | 20 to 30 | 241 | 5.24 | | | 50 | 3.57 | -1.67 | |
| | 30 to 40 | 127 | 2.76 | | | 34 | 2.43 | -0.33 | |
| | 40 to 50 | 54 | 1.17 | | | 18 | 1.29 | 0.11 | |





| 50 to 60 | 27 | 0.59 | 11 | 0.79 | 0.20 |
| 60 to 70 | 38 | 0.83 | 10 | 0.71 | -0.11 |
| 70 to 80 | 16 | 0.35 | 4 | 0.29 | -0.06 |
| 80 to 90 | 24 | 0.52 | 5 | 0.36 | -0.16 |
| 90 to 100 | 5 | 0.11 | 3 | 0.21 | 0.11 |
| 100 to 110 | 5 | 0.11 | 1 | 0.07 | -0.04 |
| 110 to 120 | 5 | 0.11 | 0 | 0.00 | -0.11 |
| 120 to 130 | 4 | 0.09 | 1 | 0.07 | -0.02 |
| 130 to 140 | 5 | 0.11 | 0 | 0.00 | -0.11 |

**Code availability**

The processing of the historical aerial images into point clouds (and orthophotos) was done with the commercial software Agisoft Metashape Professional (Version 1.6.6). These point clouds as well as the point clouds based on LiDAR data (ALS) were further processed in the commercial geoinformation system SAGA LIS Pro 3D (Version 7.4.0) and converted into DEMs. The preparatory steps for the regression lines (derivation of the corresponding value pairs (SY and SCA)) were carried out in open-source software SAGA GIS (Version 7.2.0), whereby the subsequent automated repetition of the extraction of the value pairs by using a for-loop and the calculation of the corresponding regression lines were carried out in the open-source software R (RStudio, version 1.4.1103). Maps were created in both the open-source software SAGA GIS and QGIS (Version 3.22.4). Atmospheric simulation was performed using the Advanced Research version of the Weather Research and Forecasting (ARW-WRF) model (version 4.3). The meteorological analyses were carried out in R.

**Data availability**

The historical aerial images (HAI) and the corresponding calibration certificates (if available) were provided by the Federal Office of Metrology and Surveying (BEV, Vienna, Austria) (aerial image series 1953 and 1954), by the Italian Military Geographic Institute (IGMI, Florence, Italy) (aerial image series 1945 and 1959) and by the Province of Tyrol (aerial image series 1970, 1971 and 1973). The DEM 2006 (Horlachtal) and the point clouds of 2006 and 2004/2005 (Kaunertal and Martelltal) were provided by the Province of Tyrol and the Autonomous Province of Bolzano. The historical maps of 1886/1887 (Kaunertal), 1889 (Horlachtal), 1918 (Martelltal) and 1922 (Kaunertal) were provided by the Archive of the German Alpine Club (DAV), the Ötztal Gedächtnisspeicher (Längenfeld, Austria), the BEV and the Bavarian Academy of Sciences and Humanities. The orthophotos of 2020 (all valleys) were made available for download by the Province of Tyrol and the Autonomous Province of Bolzano on their respective websites. The large-scale elevation data (DSM and Hillshade) (Overview Alps, Figure 1) was provided by Copernicus (Copernicus Land Monitoring Service). These data were produced with the financial support of the European Union. The 20th century NOAA/CIRES/DOE reanalysis data (V3) were provided by NOAA PSL, Boulder, Colorado, USA, from their website https://psl.noaa.gov. Support for the Twentieth Century Reanalysis Project version 3 dataset is provided by the U.S. Department of Energy, Office of Science Biological and Environmental Research (BER), by the National Oceanic and Atmospheric Administration Climate Program Office, and by the NOAA Earth System



Research Laboratory Physical Sciences Laboratory.⊿8364; The ERA5 dataset we used is a Copernicus product. It contains
processed information from the Copernicus Climate Change Service [2021] and the Copernicus Atmosphere Monitoring
Service [2021]. Please note that neither the European Commission nor the European Centre for Medium-Range Weather
Forecasts is responsible for any use that may be made of the Copernicus information or data contained therein.

**Author contribution**

The study was conceptualised by MA, FH, TH and MB. Data preparation was carried out by MA, JR, FF, FH, LP, MP, MW,
LB, MS and SB-N. The methodological approach was developed by MA, JR, FH and TH for the SCA modelling and MA, FH
and MP for the meteorological analysis. The formal analysis was carried out by MA and MP. Supervision was carried out by
FH, TH and MB. The original draft was prepared by MA. JR, FF, FH, TH, LP, MP, MW, LB, MS, SB-N and MB were
involved in the revision of the manuscript. MB, FH and TH were responsible for fundraising and project management.

**Competing interests**

The authors declare that they have no conflict of interest.

**Acknowledgement**

We would like to thank the German Research Foundation (DFG), the Austrian Science Fund (FWF) and the Swiss National
Science Foundation (SNF) for financial support of the research project SEHAG (SEnsitivity of High Alpine Geosystems to
climate change since 1850), within the framework of which this study was generated. Furthermore, we would like to thank for
providing the aerial images and the calibration certificates. In this context, we would like to thank the BEV, IGMI, the Province
of Tyrol, the Province of Bolzano and the Hydrographic Office of the Autonomous Province of Bolzano (Civil Protection
Agency). In addition, we would also like to thank the Province of Tyrol for providing the DEM 2006 (Horlachtal) as well as
the point cloud 2006 (Kaunertal) and the Autonomous Province of Bolzano for the point cloud 2006 (Martelltal). We would
also like to thank the Archive of the German Alpine Club (DAV), the Ötztaler Gedächtnisspeicher, the BEV and the Bavarian
Academy of Sciences and Humanities for providing the historical maps 1886/1887 (Kaunertal), 1889 (Horlachtal), 1918
(Martelltal) and 1922 (Kaunertal). We would also like to thank the Province of Tyrol and the Autonomous Province of Bolzano
for the orthophotos of 2020, which can be downloaded on their websites quickly and easily. We would also like to thank
Copernicus (Copernicus Land Monitoring Service) for the available download of the coarse resolution hillshade (Overview
Alps, Figure 1). Additionally, we would like to thank Wucher Helikopter GmbH (Ludesch, Austria) for carrying out the flights
in which the LiDAR data (ALS) 2017 (Kaunertal) and 2019 (Horlachtal and Martelltal) were acquired. Many thanks for the



safe flights even in difficult high alpine terrain. In addition, many thanks to NOAA PSL, Boulder, Colorado, USA for the
20CRv3 and Copernicus for the ERA5 dataset.
**Funding**
The study was financially supported by the German Research Foundation (DFG) and the Austrian Science Fund (FWF) (grant
numbers: BE 1118/38-1, BE 1118/39-1, BE 1118/40-1, HA 5740/10-1, HE 5747/6-1, MA 6966/4-1, LA 4426/1-1 and 4062-
N29). The open access publication of this article was supported by the Open Access Fund of the Catholic University of
Eichstätt-Ingolstadt.

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
