# Peer review of "Long-term monitoring (1953-2019) of geomorphologically active"

_EGUsphere, 2022_

## Author Comment (AC1)

**Manuskript**: Long-term monitoring (1953–2019) of geomorphologically active sections on LIA lateral moraines under changing meteorological conditions, Moritz Altmann, Madlene Pfeiffer, Florian Haas, Jakob Rom, Fabian Fleischer, Tobias Heckmann, Livia Piermattei, Michael Wimmer, Lukas Braun, Manuel Stark, Sarah Betz-Nutz, and Michael Becht
Submitted on 27 Dec 2022
**Handling editor**: Giulia Sofia, giulia.sofia@uconn.edu

**RC1**: 'Comment on egusphere-2022-1512', Anonymous Referee #1, 23 May 2023

Reviewer: The preprint of an ESurf manuscript open for interactive discussion constitutes an interesting and important study and addresses one of the important research geomorphology currently focuses on. Among many aspects of modern and future 'Global Environmental Change', the development of recently deglaciated glacier forelands and connected morphodynamic processes in mountain regions is surely of considerable significance. Despite the related concept of the 'paraglacial period' is now well established for several decades, existing work often suffers from the lack of detailed long-term observations. Some conceptual facets and underlying assumptions would certainly highly benefit from such data.

Gully formation on the slopes of lateral moraines exposed by successive glacier retreat since the 'Little Ice Age' maximum extension is a prominent example of paraglacial processes and characteristic for many glacier forelands worldwide. The current study presents such detailed and accurate data from a total of twelve active gully systems on lateral moraines in five glacier forelands, all located within the Eastern European Alps. Several surveys over a rather long timeframe from 1953 to 2019 allow a detailed investigation of gully system development and sediment yield over a comparable long time and 3 separate time periods within this interval. The authors apply a highly developed methodology related to both the acquisition of aerial imagery, DEMs, LiDAR scenes etc. and their subsequent morphometric analysis. All results are well presented and all data are of high quality and accuracy. All individual methodological steps within the data analysis and well explained alongside all necessary information on the data base. It is, therefore, no surprise that the results of the truly long-term study are impressive and a valuable contribution to the topic.

In many aspects, the majority of the investigated sites confirm the validity of the established 'paraglacial period concept', for example with the highlighted decrease of sediment yield over time during the targeted time period (i.e. the 'sediment-exhaustion' model). On the other hand, an interesting result is that the gullies have still not been fully stabilised and some morphodynamic activity is still recorded during the final (youngest) time period. This is in disagreement to some existing studies, but also in agreement with other ones. Alongside a newly developed 'sediment activity concept' this finding and two sites that do, as exceptions among the total data set, not fully follow the expected conceptual decreasing sediment yield for 'paraglacial processes' are finally discussed. But here, a deeper discussion on basis of published work from other regions would likely improve the high quality of the manuscript. The new concept is introduced and discussed too briefly to convince that the observations justify the introduction of a new concept. An extension of the discussion chapter should serve to interpret the results of the study in more detail and highlight the differences from the established concept and assumption to underline its significance.

It seems only a minor points given the scientific value of the preprint, but an excessive use of acronyms and abbreviations for terms where they are neither necessary nor established negatively affects fluent reading, to an extent where it is annoying. It would be acceptable for a technical report, but for a journal article this should be avoided. Because it surely constitutes

no major effort to make related changes in the text alongside some final editorial and typographic changes, the authors are recommended to consider such changes (some examples are indicated in the technical comment section below).

Summarising, the long-term study and its well presented results constitute a valuable contribution to a wider audience within the targeted scientific community. The only scientific room for improvement is a recommended extension of the discussion section by adding some depths. In my specific comments below I will address this in more detail.

**Specific comments:**

I feel that the authors should extent the discussion chapter by exploring some of their most interesting findings, for example that their investigated active gully systems still are active and show, despite a decrease of sediment yield in most cases, no stabilisation. As correctly stated, this finding is different to previously published work from other regions, for example Western Norway. Perhaps the authors should present a hypothesis or some possible reasons for this, simply because it is to some extent contradictors to the established sediment-exhaustion concept for the development of gullies. Regional different conditions of gully development need, however, to be taken into account with the discussion of this apparent discrepancy. Among those are the different geomorphological setting (typical Alpine-type lateral moraines vs. debris-covered slopes of different origin in Norway) and the sedimentological properties of the lateral moraines related to their genetic origin etc. Factor other than the morphometric properties have to be taken into account.

Authors: Hypotheses or possible reasons are required as to why geomorphic active areas on LIA lateral moraines stabilise over several decades in some regions and not in others. This is difficult to determine and cannot be finally investigated or clarified. It is assumed that this is due to local conditions. This could be due to the changing triggering events caused by changing heavy rainfall events, also to the different sedimentological properties of the lateral moraines in relation to the genetic origin or the different geomorphological settings. Furthermore, it could be due to the different characteristics of the lateral moraine sections, such as slope gradient, slope length, time of ice exposure and the different development of vegetation. It would be possible to include this in the discussion, but ultimately only assumptions can be made.

Reviewer: The 'sediment activity concept' developed by the authors is only comparatively briefly introduced. With a limited number of study sites and - as least this is my (potentially wrong) assumption - mainly based of two exemptions from the trend the basis for developing such an innovative concept is rather small. And with the 'ice release' not included and a limited temporal validity (see lines 556 ff.) the authors need to properly elaborate is this constitutes a significant new and valuable concept - in other words justify that their observations support such a step instead of accept that exceptions from other established concepts always may exist. Perhaps it would strengthen the value and depths of the discussion chapter if the authors focus more on the investigation of potential reasons for the deviating date of these two gully systems instead of developing a new and obviously limited concept.

Authors: We would like to keep the development of the sediment activity concept to the number of study areas in this study. However, what we can include in the discussion are the results of this study and the developed concept with the description of the development of the lateral moraines of other studies. Thus, there is a broader discussion with already existing results. The concept is indeed well compatible with already existing studies (e.g. Betz-Nutz et

al. (2023)), as geomorphologically active areas could also be identified there, which on the one hand show a decreasing trend, have activity at similar levels or show an increase. Reasons for this different development can also be brought into the discussion, which are based on the fact that each slope has very individual conditions that can also lead to an increase in geomorphological activity over several decades.

(Betz-Nutz, S., Heckmann, T., Haas, F., and Becht, M.: Development of the morphodynamics on Little Ice Age lateral moraines in 10 glacier forefields of the Eastern Alps since the 1950s, Earth Surf. Dynam., 11, 203–226, https://doi.org/10.5194/esurf-11-203-2023, 2023.).

Reviewer: In their discussion section 5.4 'meteorological drivers' the authors present an interesting review on potential meteorological drivers for the current morphodynamic activity of the gully systems investigated. In contrast to what some readers may have expected based on frequently emphasised (popular)scientific statements, there seems to be major increase in the frequency or magnitude of heavy-precipitation events in the study areas. They accurately describe the differences of simulated vs. observed meteorological data what is good and provides good insights. But as this aspect of the study is even included in the title, it seems that some summarising conclusion (or assumptions) regarding the potential influence on climate change nerd to be provided. These could be along the lines 'paraglacial period'/'sediment exhaustion' concept vs. development of geomorphological activity and morphodynamics in times of Global Change. And to throw in just a provocative hypothesis: Could future climate change lead to increased morphodynamic activity (erosion) and disturb the 'normal' decrease of sediment yield as predicted by the established paraglacial period concept?

Authors: No increase in the frequency and magnitude of heavy precipitation events could be demonstrated. The hypothesis proposed by the reviewer seems too strong to us. Finally, assumptions are made as to why the warming of the air temperature could have had a positive effect on erosion (increase in the number of warm-air inflows). Ultimately, however, the exact influence cannot be proven. The general decrease is best explained by the generally effecting paraglacial adjustment processes. Special cases always show special characteristics. We would therefore rather suggest changing the title to: Long-term monitoring (1953-2019) of geomorphologically active sections on LIA lateral moraines in the context of changing meteorological conditions.

Reviewer: I am confident that by investing some effort to extend and strengthen some sections of the discussion chapter will substantially increase the overall scientific value of the comprehensive and important research presented by the authors. The overall goal should be to place the significant long-term approach and its results better in a general context, also beyond the Eastern European Alps.

**Technical corrections:**

The manuscript is mostly well structured and written, with the important exception of excessive use of (to a considerable extent) unnecessary acronyms/abbreviations that can make reading a pain. A few typographic/editorial changes should, however, be addressed during the revision:

Authors: All acronyms and abbreviations are checked. The established ones are to be kept, the less established abbreviations or those developed specifically for this study are removed and not abbreviated but written out in full.

- Reviewer: I feel that in the title the acronym 'LIA', despite well established, should be written in full. Also, I think it should be '...sections of ...' and, given the time period investigated, perhaps better 'climatic conditions'.
- Authors: The proposed changes can be incorporated and the title can be changed accordingly: Long-term monitoring (1953-2019) of geomorphologically active sections of Little Ice Age lateral moraines in the context of changing climatic conditions.
- Reviewer: Line 13: Add 'European' to Alps (only once in abstract and general text).
- Authors: The proposed amendment is adopted.
- Reviewer: Line 19/20: I recommend to consider different expressions for 'areas of interests' and 'entire areas of interest'. As I understand it, this refers to the active gully system and the entire lateral moraine. Why making it unnecessarily complicated with an excessive use of the term 'area of interest'? 'Sites' or gully systems allow the reader to read the abstract more fluently
- Authors: The proposed amendment is adopted.
- Reviewer: Line 21: 'Can be shown'
- Authors: The proposed amendment is adopted.
- Reviewer: Line 21: 'Epochs' is a wrong term in the context.
- Authors: The proposed amendment is adopted. The sentence is amended accordingly: Subsequently, both the areas of interest and the different time periods of both approaches are compared.
- Reviewer: Line 21: This sentence should be re-written as it is a bit unclear.
- Authors: The sentence is amended accordingly: Based on the slopes of the calculated regression lines, it could be shown that the highest variability of sediment yield in the areas of interest occurs in the first epoch (mainly 1950s to 1970s). This can be attributed to the fact that in some areas of interest the sediment yield per square metre increases clearly more strongly (regression lines with slopes up to 1.5). In contrast, in the later epochs (1970s to mid-2000s and mid-2000s to 2017/2019), there is generally a decrease in 10 out of 12 cases (regression lines with slopes around 1).
- Reviewer: Line 34: Better: 'with influence of dead ice over decades'
- Authors: The sentence is amended accordingly.
- Reviewer: Keywords: A quite high number, are all necessary? Paraglacial process system should be added
- Authors: The keyword Modelling is removed. The keyword "paraglacial process system" is added as suggested.
- Reviewer: Line 42: There is no defined or general 'end' of the Little Ice Age. The authors could well be more specific and relate it to the Eastern European Alps (if they wish).
- Authors: The end of the Little Ice Age is also underpinned with literature (Matthews, J. A., & Briffa, K. R. (2005). The 'little ice age': Re-evaluation of an evolving concept. Geografiska Annaler. Series A. Physical Geography, 87(1), 17–36. https://doi.org/10.1111/j.0435-3676.2005.00242.x and Ivy-Ochs, S., Kerschner, H., Maisch, M., Christl, M., Kubik, P. W., & Schlüchter, C. (2009). Latest Pleistocene and Holocene glacier variations in the European Alps. Quaternary Science Reviews, 28(21-22), 2137–2149. https://doi.org/10.1016/j.quascirev.2009.03.009
- Reviewer: Line 44: Better: 'extending'
- Authors: The proposal can be accepted and amended accordingly.
- Reviewer: Line 61: Better: 'and subsequently'
- Authors: The proposal can be accepted and amended accordingly.
- Reviewer: Line 97: A recent comparative study could be worth being cited in this context: Eichel, J., Draebing, D., Winkler, S. & Meyer, N. (2023): Similar vegetationgeomorphic disturbance feedbacks shape unstable glacier forelands across mountain regions. *Ecosphere* 14(2), e4404.

- Authors: The proposal can be accepted and amended accordingly.
- Reviewer: Lines 104, 111: The inflationary use of acronyms/abbreviations in this chapter (ans subsequent ones) makes it a bit hard to read the text fluently. Whereas for very established and complex term (DEM, SfM, LiDAR) it is all fine, abbreviating 'historical aerial imagery' and in particular 'sediment yield' goes over the top. The space saved does not compensate for poor readability with so many acronyms.
- Authors: The suggestion can be accepted and amended accordingly.
- Reviewer: Line 124: Explain the acronym AOI the first time it is used in the general text - or much better avoid this term at all.
- Authors: The suggestion can be accepted and amended accordingly.
- Reviewer: Line 124: See comment to 'epoch' above, why not 'period' or 'time period'?
- Authors: For better understanding, the term epoch is changed to time period.
- Reviewer: Line 128: Add 'European'
- Authors: Will be added.
- Reviewer: Line 138: Capital letters for 'Main Alpine Divide'
- Authors: Will be added.
- Reviewer: Line 153: 'Sparse' for 'low'
- Authors: Will be changed.
- Reviewer: Figure 1: Although the term is 'Gletschervorfeld' in German, the appropriate term is 'foreland' not 'forefield'. It would be good to (a) mention 'glacier outline' or 'glacier margin' in the legends as well. Please use the same colour for the same glacier extent (e.g. not green for the 1953 margin at Gepatschferner and blue for the same margin at Weißseeferner). Better to only use one colour for one data on all
- Authors: The suggestions will be changed. The terms foreland and glacier outline are amended. Furthermore, the same colour of the glacier stands should be chosen, which can be assumed.
- Reviewer: Line 180: Is it necessary to give such a detailed information? The same applies for the full project titles on Table 3.
- Authors: This section can be shortened, but we would like to have some information about the ALS recordings and previous projects.
- Reviewer: Line 245 ff.: This sentence could be made clearer by re-wording it.
- Authors: The sentence will be rewritten accordingly: No threshold has been set for the level of detection of the DoDs, as Anderson (2019) clearly recommends not using this for volumetric calculations as it leads to bias in the results (Anderson, S. W.: Uncertainty in quantitative analyses of topographic change: error propagation and the role of thresholding, Earth Surf. Process. Landforms, 44, 1015–1033, https://doi.org/10.1002/esp.4551, 2019.).
- Reviewer: Line 365/366: Find a different solution for the two subsequent brackets and re-word the sentence
- Authors: The sentence will be rewritten accordingly: With regard to section 3.3, a decreasing intercept together with an almost constant, although slightly decreasing, slope close to one can be seen over the different epochs in the log-log model, indicating that the relation between SCA and SY remains almost constant.
- Reviewer: Line 368/369: This sentence is a good example what I highlighted as excessive use of acronyms. Apart from those for the different active gully sections in the sentence above, no other acronym is necessary here, neither AOI nor SCA or SY.
- Authors: This will be amended as suggested.
- Reviewer: Line 501/502: If the authors would like to find a citation for this assumption, they could use [Jäger, D. & Winkler, S. (2012): Paraglacial processes on

the glacier foreland of Vernagtferner (Ötztal Alps, Austria). *Zeitschrift für Geomorphologie N.F. Supplement Bd.* 56 (4): 95 – 113.] where this influence is described from another glacial foreland in the region.

- Authors: The reference suggestion has been read and will be added.

---

## Author Response (AR1)

**Manuskript**: Long-term monitoring (1953–2019) of geomorphologically active sections on LIA lateral moraines under changing meteorological conditions, Moritz Altmann, Madlene Pfeiffer, Florian Haas, Jakob Rom, Fabian Fleischer, Tobias Heckmann, Livia Piermattei, Michael Wimmer, Lukas Braun, Manuel Stark, Sarah Betz-Nutz, and Michael Becht
Submitted on 27 Dec 2022
**Handling editor**: Giulia Sofia, giulia.sofia@uconn.edu

**RC1**: 'Comment on egusphere-2022-1512', Anonymous Referee #1, 23 May 2023

Reviewer: The preprint of an ESurf manuscript open for interactive discussion constitutes an interesting and important study and addresses one of the important research geomorphology currently focuses on. Among many aspects of modern and future 'Global Environmental Change', the development of recently deglaciated glacier forelands and connected morphodynamic processes in mountain regions is surely of considerable significance. Despite the related concept of the 'paraglacial period' is now well established for several decades, existing work often suffers from the lack of detailed long-term observations. Some conceptual facets and underlying assumptions would certainly highly benefit from such data.

Gully formation on the slopes of lateral moraines exposed by successive glacier retreat since the 'Little Ice Age' maximum extension is a prominent example of paraglacial processes and characteristic for many glacier forelands worldwide. The current study presents such detailed and accurate data from a total of twelve active gully systems on lateral moraines in five glacier forelands, all located within the Eastern European Alps. Several surveys over a rather long timeframe from 1953 to 2019 allow a detailed investigation of gully system development and sediment yield over a comparable long time and 3 separate time periods within this interval. The authors apply a highly developed methodology related to both the acquisition of aerial imagery, DEMs, LiDAR scenes etc. and their subsequent morphometric analysis. All results are well presented and all data are of high quality and accuracy. All individual methodological steps within the data analysis and well explained alongside all necessary information on the data base. It is, therefore, no surprise that the results of the truly long-term study are impressive and a valuable contribution to the topic.

In many aspects, the majority of the investigated sites confirm the validity of the established 'paraglacial period concept', for example with the highlighted decrease of sediment yield over time during the targeted time period (i.e. the 'sediment-exhaustion' model). On the other hand, an interesting result is that the gullies have still not been fully stabilised and some morphodynamic activity is still recorded during the final (youngest) time period. This is in disagreement to some existing studies, but also in agreement with other ones. Alongside a newly developed 'sediment activity concept' this finding and two sites that do, as exceptions among the total data set, not fully follow the expected conceptual decreasing sediment yield for 'paraglacial processes' are finally discussed. But here, a deeper discussion on basis of published work from other regions would likely improve the high quality of the manuscript. The new concept is introduced and discussed too briefly to convince that the observations justify the introduction of a new concept. An extension of the discussion chapter should serve to interpret the results of the study in more detail and highlight the differences from the established concept and assumption to underline its significance.

It seems only a minor points given the scientific value of the preprint, but an excessive use of acronyms and abbreviations for terms where they are neither necessary nor established negatively affects fluent reading, to an extent where it is annoying. It would be acceptable for a technical report, but for a journal article this should be avoided. Because it surely constitutes

no major effort to make related changes in the text alongside some final editorial and typographic changes, the authors are recommended to consider such changes (some examples are indicated in the technical comment section below).

Summarising, the long-term study and its well presented results constitute a valuable contribution to a wider audience within the targeted scientific community. The only scientific room for improvement is a recommended extension of the discussion section by adding some depths. In my specific comments below I will address this in more detail.

**Specific comments:**

I feel that the authors should extent the discussion chapter by exploring some of their most interesting findings, for example that their investigated active gully systems still are active and show, despite a decrease of sediment yield in most cases, no stabilisation. As correctly stated, this finding is different to previously published work from other regions, for example Western Norway. Perhaps the authors should present a hypothesis or some possible reasons for this, simply because it is to some extent contradictors to the established sediment-exhaustion concept for the development of gullies. Regional different conditions of gully development need, however, to be taken into account with the discussion of this apparent discrepancy. Among those are the different geomorphological setting (typical Alpine-type lateral moraines vs. debris-covered slopes of different origin in Norway) and the sedimentological properties of the lateral moraines related to their genetic origin etc. Factor other than the morphometric properties have to be taken into account.

Authors: Hypotheses or possible reasons are required as to why geomorphic active areas on LIA lateral moraines stabilise over several decades in some regions and not in others. This is difficult to determine and cannot be finally investigated or clarified. It is assumed that this is due to local conditions. This could be due to the changing triggering events caused by changing heavy rainfall events, also to the different sedimentological properties of the lateral moraines in relation to the genetic origin or the different geomorphological settings. Furthermore, it could be due to the different characteristics of the lateral moraine sections, such as slope gradient, slope length, time of ice exposure and the different development of vegetation. It would be possible to include this in the discussion, but ultimately only assumptions can be made.

Therefore, the following section was added to the discussion (Sec. 5.3.2 Erosion rates):

Comparing the long-term erosion rates of gully systems from different studies ultimately shows high variability in the adjustment; as these studies were also conducted using different methods, on different time scales, and in different glacier forelands. Differences are probably mainly due to the different local conditions, such as the geomorphological settings, e.g. the different characteristics of the lateral moraine sections, such as slope gradient, slope length, time of ice exposure, dead ice influence and the development of vegetation. Furthermore, the lateral moraines have different sedimentological characteristics related to their genetic origin. In addition, different meteorological conditions prevail in the different regions.

Reviewer: The 'sediment activity concept' developed by the authors is only comparatively briefly introduced. With a limited number of study sites and - as least this is my (potentially wrong) assumption - mainly based of two exemptions from the trend the basis for developing such an innovative concept is rather small. And with the 'ice release' not included and a limited temporal validity (see lines 556 ff.) the authors need to properly elaborate is this

constitutes a significant new and valuable concept - in other words justify that their observations support such a step instead of accept that exceptions from other established concepts always may exist. Perhaps it would strengthen the value and depths of the discussion chapter if the authors focus more on the investigation of potential reasons for the deviating date of these two gully systems instead of developing a new and obviously limited concept.

Authors: We would like to keep the development of the „Sediment activity concept" to the number of test sites in this study. However, what we can additionally include in the discussion are the results of other studies. This will result in a broader discussion with existing results. The concept is indeed well compatible with already existing studies, e.g., Betz-Nutz et al. (2023), since geomorphologically active areas could also be identified there, which on the one hand show a decreasing trend, have activity at similar levels, or show an increase. Similarly, the study by Church and Ryder (1972), Ballantyne (2002), Ballantyne & Benn (1996), Curry et al. (2006), Curry (1999) and Schiefer and Gilbert (2007) show a decrease in geomorphological activity, which is equally consistent with the „Sediment activity concept" presented here. The possible reasons for the divergent developments of different lateral moraine sections has already been sufficiently discussed.

This section was therefore incorporated into the discussion (Sec. 5.3.1 „The sediment activity concept"):

The "Sediment activity concept " presented here is also compatible with the results of other studies, as for example Betz-Nutz et al. (2023) mostly show a decrease in erosion rates over a similar time period, but also a remaining at similar levels and an increase of the morphodynamics. In addition, the studies by Church and Ryder (1972), Ballantyne and Benn (1996), Curry (1999), Ballantyne (2002a), Curry et al. (2006) and Schiefer and Gilbert (2007) show a decrease in geomorphic activity over time, which is also consistent with the model presented here.

- Ballantyne, C.K., 2002. Paraglacial geomorphology. Quaternary Science Reviews 21, 1935-2017.
- Ballantyne, C.K., Benn, D.I., 1996. Paraglacial slope adjustment during recent deglaciation and its implications for slope evolution in formerly glaciated environments. Anderson, M. G., & Brooks, S. M. (eds.), Advances in hillslope processes 2, 1173-1195.
- Betz-Nutz, S., Heckmann, T., Haas, F., Becht, M., 2023. Development of the morphodynamics on Little Ice Age lateral moraines in 10 glacier forefields of the Eastern Alps since the 1950s. Earth Surf. Dynam. 11, 203-226.
- Church, M., Ryder, J.M., 1972. Paraglacial Sedimentation: A Consideration of Fluvial Processes Conditioned by Glaciation. Geol Soc America Bull 83, 3059.
- Curry, A.M., 1999. Paraglacial modification of slope form. Earth Surf. Process. Landforms 24, 1213-1228.
- Curry, A.M., Cleasby, V., Zukowskyj, P., 2006. Paraglacial response of steep, sediment-mantled slopes to post-'Little Ice Age' glacier recession in the central Swiss Alps. J. Quaternary Sci. 21, 211-225.
- Schiefer, E., Gilbert, R., 2007. Reconstructing morphometric change in a proglacial landscape using historical aerial photography and automated DEM generation. Geomorphology 88, 167-178.

Reviewer: In their discussion section 5.4 'meteorological drivers' the authors present an interesting review on potential meteorological drivers for the current morphodynamic activity

of the gully systems investigated. In contrast to what some readers may have expected based on frequently emphasised (popular)scientific statements, there seems to be major increase in the frequency or magnitude of heavy-precipitation events in the study areas. They accurately describe the differences of simulated vs. observed meteorological data what is good and provides good insights. But as this aspect of the study is even included in the title, it seems that some summarising conclusion (or assumptions) regarding the potential influence on climate change nerd to be provided. These could be along the lines 'paraglacial period'/'sediment exhaustion' concept vs. development of geomorphological activity and morphodynamics in times of Global Change. And to throw in just a provocative hypothesis: Could future climate change lead to increased morphodynamic activity (erosion) and disturb the 'normal' decrease of sediment yield as predicted by the established paraglacial period concept?

Authors: No increase in the frequency and magnitude of heavy precipitation events could be demonstrated. The hypothesis proposed by the reviewer seems too strong to us. Finally, assumptions are made as to why the warming of the air temperature could have had a positive effect on erosion (increase in the number of warm-air inflows). Ultimately, however, the exact influence cannot be proven. The general decrease is best explained by the generally effecting paraglacial adjustment processes. Special cases always show special characteristics. We would therefore rather suggest changing the title to: Long-term monitoring (1953-2019) of geomorphologically active sections of Little Ice Age lateral moraines in the context of changing meteorological conditions.

Reviewer: I am confident that by investing some effort to extend and strengthen some sections of the discussion chapter will substantially increase the overall scientific value of the comprehensive and important research presented by the authors. The overall goal should be to place the significant long-term approach and its results better in a general context, also beyond the Eastern European Alps.

**Technical corrections:**

The manuscript is mostly well structured and written, with the important exception of excessive use of (to a considerable extent) unnecessary acronyms/abbreviations that can make reading a pain. A few typographic/editorial changes should, however, be addressed during the revision:

Authors: All acronyms and abbreviations are checked. The established ones are to be kept (DEM, DoD, ALS, LIA, SfM), the less established abbreviations or those developed specifically for this study are removed and not abbreviated but written out in full. The following were written out: AOI, SCA and SY.

- Reviewer: I feel that in the title the acronym 'LIA', despite well established, should be written in full. Also, I think it should be '...sections of ...' and, given the time period investigated, perhaps better 'climatic conditions'.
- Authors: The proposed changes are incorporated and the title is changed accordingly: Long-term monitoring (1953-2019) of geomorphologically active sections of Little Ice Age lateral moraines in the context of changing climatic conditions.
- Reviewer: Line 13: Add 'European' to Alps (only once in abstract and general text).
- Authors: The proposed amendment was adopted. Also in the lines: 127/128, 137/138, 598, 695, 725

- Reviewer: Line 19/20: I recommend to consider different expressions for 'areas of interests' and 'entire areas of interest'. As I understand it, this refers to the active gully system and the entire lateral moraine. Why making it unnecessarily complicated with an excessive use of the term 'area of interest'? 'Sites' or gully systems allow the reader to read the abstract more fluently
- Authors: The proposed amendment was adopted. We used the term sites.
- Reviewer: Line 21: 'Can be shown'
- Authors: The proposed amendment was adopted (Could be shown –> Can be shown).
- Reviewer: Line 21: 'Epochs' is a wrong term in the context.
- Authors: The proposed amendment is adopted. The sentence is amended accordingly: Subsequently, both the sites and the different time periods of both approaches are compared.
- Reviewer: Line 21: This sentence should be re-written as it is a bit unclear.
- Authors: The sentence was amended accordingly: Based on the slopes of the calculated regression lines, it can be shown that the highest variability of sediment yield in the sites occurs in the first epoch (mainly 1950s to 1970s). This can be attributed to the fact that within some sites the sediment yield per square metre increases clearly more strongly (regression lines with slopes up to 1.5). In contrast, in the later epochs (1970s to mid-2000s and mid-2000s to 2017/2019), there is generally a decrease in 10 out of 12 cases (regression lines with slopes around 1).
- Reviewer: Line 34: Better: 'with influence of dead ice over decades'
- Authors: The sentence was amended accordingly.
- Reviewer: Keywords: A quite high number, are all necessary? Paraglacial process system should be added
- Authors: The keyword Modelling was removed. The keyword "paraglacial process system" was added as suggested.
- Reviewer: Line 42: There is no defined or general 'end' of the Little Ice Age. The authors could well be more specific and relate it to the Eastern European Alps (if they wish).
- Authors: The end of the Little Ice Age was also underpinned with literature. So that it is better classified now:

  (1) Matthews, J. A., & Briffa, K. R. (2005). The 'little ice age': Re-evaluation of an evolving concept. Geografiska Annaler. Series A. Physical Geography, 87(1), 17–36. https://doi.org/10.1111/j.0435-3676.2005.00242.x
  (2) Ivy-Ochs, S., Kerschner, H., Maisch, M., Christl, M., Kubik, P. W., & Schlüchter, C. (2009). Latest Pleistocene and Holocene glacier variations in the European Alps. Quaternary Science Reviews, 28(21-22), 2137–2149. https://doi.org/10.1016/j.quascirev.2009.03.009

  Reviewer: Line 44: Better: 'extending'

- Authors: The proposal was amended.
- Reviewer: Line 61: Better: 'and subsequently'
- Authors: The proposal was accepted and amended accordingly.
- Reviewer: Line 97: A recent comparative study could be worth being cited in this context: Eichel, J., Draebing, D., Winkler, S. & Meyer, N. (2023): Similar vegetation-geomorphic disturbance feedbacks shape unstable glacier forelands across mountain regions. *Ecosphere* 14(2), e4404.
- Authors: The proposal was accepted and amended accordingly.

- Reviewer: Lines 104, 111: The inflationary use of acronyms/abbreviations in this chapter (ans subsequent ones) makes it a bit hard to read the text fluently. Whereas for very established and complex term (DEM, SfM, LiDAR) it is all fine, abbreviating 'historical aerial imagery' and in particular 'sediment yield' goes over the top. The space saved does not compensate for poor readability with so many acronyms.
- Authors: The suggestion was accepted and modified accordingly.
- Reviewer: Line 124: Explain the acronym AOI the first time it is used in the general text - or much better avoid this term at all.
- Authors: The suggestion was accepted and changed accordingly. The term was avoided and changed to sites (in this case). AOI is also avoided, replaced by site.
- Reviewer: Line 124: See comment to 'epoch' above, why not 'period' or 'time period'?
- Authors: For better understanding, the term epoch was changed to time period.
- Reviewer: Line 128: Add 'European'
- Authors: Was added.
- Reviewer: Line 138: Capital letters for 'Main Alpine Divide'
- Authors: Was added.
- Reviewer: Line 153: 'Sparse' for 'low'
- Authors: Was changed.
- Reviewer: Figure 1: Although the term is 'Gletschervorfeld' in German, the appropriate term is 'foreland' not 'forefield'. It would be good to (a) mention 'glacier outline' or 'glacier margin' in the legends as well. Please use the same colour for the same glacier extent (e.g. not green for the 1953 margin at Gepatschferner and blue for the same margin at Weißseeferner). Better to only use one colour for one data on all
- Authors: The proposals were changed. The terms foreland and glacier outline were changed. Also, the same color of the glacier outlines should be chosen, which was accepted and changed accordingly.
- Reviewer: Line 180: Is it necessary to give such a detailed information? The same applies for the full project titles on Table 3.
- Authors: This section was shortened, but we would like to have some detailed information about the ALS recordings and previous projects. The following section has been removed from the manuscript: Subsequently, the GNSS/IMU trajectory data were processed in three steps. This included, (i) the calculation of precise trajectories using the software PosPac MMS (Applanix), (ii) the attachment of raw scans to the flight lines using the software package Riegl RiProcess, and finally (iii) a strip adjustment in the processing software OPALS (Pfeifer et al., 2014) using the approach of Glira et al. (2015).

(1) Pfeifer, N., Mandlburger, G., Otepka, J., and Karel, W.: OPALS – A framework for Airborne Laser Scanning data analysis, Computers, Environment and Urban Systems, 45, 125–136, https://doi.org/10.1016/j.compenvurbsys.2013.11.002, available at: http://dx.doi.org/10.1016/j.compenvurbsys.2013.11.002, 2014
(2) Glira, P., Pfeifer, N., Briese, C., and Ressl, C.: A Correspondence Framework for ALS Strip Adjustments based on Variants of the ICP Algorithm, 275–289, https://doi.org/10.1127/PFG/2015/0270, 2015.

- Reviewer: Line 245 ff.: This sentence could be made clearer by re-wording it.
- Authors: The sentence was rewritten accordingly: No threshold has been set for the level of detection of the DoDs, as Anderson (2019) clearly recommends not using this for volumetric calculations as it leads to bias in the results.

- Anderson, S. W.: Uncertainty in quantitative analyses of topographic change: error propagation and the role of thresholding, Earth Surf. Process. Landforms, 44, 1015–1033, https://doi.org/10.1002/esp.4551, 2019.
- Reviewer: Line 365/366: Find a different solution for the two subsequent brackets and re-word the sentence
- Authors: The sentence was rewritten accordingly: With regard to section 3.3, a decreasing intercept together with an almost constant, although slightly decreasing, slope close to one can be seen over the different time periods in the log-log model, indicating that the relation between SCA and sediment yield remains almost constant.
- Reviewer: Line 368/369: This sentence is a good example what I highlighted as excessive use of acronyms. Apart from those for the different active gully sections in the sentence above, no other acronym is necessary here, neither AOI nor SCA or SY.
- Authors: This was changed as suggested.
- Reviewer: Line 501/502: If the authors would like to find a citation for this assumption, they could use [Jäger, D. & Winkler, S. (2012): Paraglacial processes on the glacier foreland of Vernagtferner (Ötztal Alps, Austria). *Zeitschrift für Geomorphologie N.F. Supplement Bd.* 56 (4): 95 – 113.] where this influence is described from another glacial foreland in the region.
- Authors: The reference suggestion has been read and was accepted/added.

**Manuskript**: Long-term monitoring (1953–2019) of geomorphologically active sections on LIA lateral moraines under changing meteorological conditions

Moritz Altmann, Madlene Pfeiffer, Florian Haas, Jakob Rom, Fabian Fleischer, Tobias Heckmann, Livia Piermattei, Michael Wimmer, Lukas Braun, Manuel Stark, Sarah Betz-Nutz, and Michael Becht

Submitted on 27 Dec 2022

**Handling editor**: Giulia Sofia, giulia.sofia@uconn.edu

**RC2**: 'Comment on egusphere-2022-1512', Anonymous Referee #2, 22 Jun 2023

Reviewer: This study analyzes the complex long-term geo-morphological dynamics (about 60 years) that took place in in five glacier forelands within the Eastern European Alps, by means of a data-analysis approach. Specifically, the main challenge of the work, if I understood correctly, is to derive a procedure based on acquisition of aerial imagery, DEMs, LiDAR scenes, laser scanner images, etc., to monitor the sediment yield and the volume production over a time horizon of 6 decades. Analyses of the meteorological drivers are also added.

The work is within the scope of the journal, and may provide an important contribution to the specific field of geomorphological active gully systems on Little Ice Age, although other similar efforts were published by the same authors for the same area (see Altmann et al., 2020 or Betz-Nutz et al., 2023).

I argue the choice of the title 'under changing meteorological conditions' for the reasons I will explain in the following. Additionally, I found the reading particularly tough and, to my opinion, the clarity of the presentation can be improved.

Please read in the following my general and specific observations.

General observations:

- My main concerns regard the analyses of the meteorological drivers. (i) WRF simulation data are normally used to make forecasts. I don't' see the need to use such data here, unless justified.

  Authors: WRF has been successfully employed in simulations of past climate over different areas of the Earth. Some examples of studies using WRF to simulate the past climate are: Collier et al. (2015), Collier and Immerzeel (2015), Pieri et al. (2015), Pontoppidan et al. (2016), Collier et al. (2017), Warscher et al. (2019), Collier and Mölg (2020), Wang et al. (2020).

  Collier, E., and Immerzeel, W. W.: High-resolution modeling of atmospheric dynamics in the Nepalese Himalaya. Journal of Geophysical Research: Atmospheres, 120(19), 9882-9896, DOI:10.1002/2015JD023266 , 2015.

  Collier, E., Maussion, F., Nicholson, L., Mölg, T., Immerzeel, W. W., and Bush, A. B. G.: Impact of debris cover on glacier ablation and atmosphere–glacier feedbacks in the Karakoram, The Cryosphere, 9, 1617-1632, https://doi.org/10.5194/tc-9-1617-2015 , 2015.

Collier, E., Mölg, T., and Sauter, T.: Recent atmospheric variability at Kibo summit, Kilimanjaro, and its relation to climate mode activity, J. Climate, 31, 3875–3891, https://doi.org/10.1175/JCLI-D-17-0551.1 , 2018.

Collier, E., and Mölg, T.: BAYWRF: a high-resolution present-day climatological atmospheric dataset for Bavaria, Earth Syst. Sci. Data, 12, 3097–3112, https://doi.org/10.5194/essd-12-3097-2020, 2020.

Pieri, A. B., von Hardenberg, J., Parodi, A., and Provenzale, A.: Sensitivity of precipitation statistics to resolution, microphysics, and convective parameterization: A case study with the high-resolution WRF climate model over Europe, Journal of Hydrometeorology, 16(4), 1857-1872, 2015.

Pontoppidan, M., Reuder, J., Mayer, S., & Kolstad, E. W.: Downscaling an intense precipitation event in complex terrain: the importance of high grid resolution. Tellus A: Dynamic Meteorology and Oceanography, 1271561, https://doi.org/10.1080/16000870.2016.1271561 , 2017.

Wang, X., Tolksdorf, V.,Otto, M., and Scherer, D.: WRF-based dynamical downscaling of ERA5 reanalysis data for High Mountain Asia: Towards a new version of the High Asia Refined analysis, Int. J. Climatol., 41, 743–762, DOI: 10.1002/joc.6686, 2020.

Warscher, M., Wagner, S., Marke, T., Laux, P., Smiatek, G., Strasser, U., and Kunstmann, H.: A 5 km Resolution Regional Climate Simulation for Central Europe: Performance in High Mountain Areas and Seasonal, Regional and Elevation-Dependent Variations, Atmosphere (Basel), 10, 682–715, https://doi.org/10.3390/atmos10110682 , 2019.

Reviewer: In fact, the use of the two climate products (from reanalysis and from WRF) for two different time horizon can be dangerous, because they are affected by different uncertainties and different bias errors, which are normally adjusted by means of bias correction method, and that can affect the retro-respective assessment.

Authors: To solve this problem, we will drop the ERA5 dataset and thus end the meteorological analysis for 2015. This means that meteorological data is missing until 2017/2019 (Analysis of geomorphological activity), but it is only one data source. We believe that the few years are expendable.

Reviewer: I would have used ground observations, given also the very small areas of interests. Please explain the choice and identify eventual meteorological stations.

Authors: In Kaunertal we have three weather stations (Gepatsch Alm station: Data since 2009, Weißsee station: Data since 2006). The aim was to go back to the 1950s, so there are no ground observations.

Reviewer: (ii) In fact, an additional concern regards the spatial resolution of the climate model products, that are characterized by a large resolution. Which is the overlap with the areas of interest? Provide a figure and discuss.

Authors: The required figures are shown below. The black squares represent the resolution of the simulated climate data. Climate data were extracted once per glacier foreland, whereby these were taken from the centre of the corresponding test sites.

Due to the fact that test sites were grouped together, some individual test sites are located in a different grid. Therefore, an altitude correction was made for each site where climate data were extracted.

[Figure]

Fig. 1: Spatial resolution (2x2 km) of the simulated climate data and location of the extraction of the simulated climate data in the glacier foreland of the Grastalferner (Grastal Glacier).

[Figure]

[Figure]

Fig. 3: Spatial resolution (2x2 km) of the simulated climate data and location of the extraction of the simulated climate data in the glacier foreland of the Hohenferner (Hohen Glacier).

- Reviewer: There is no real analysis of the link between morphodynamics processes and climate drivers; the study provides a screening of the meteorological series, that is fine. Therefore, the statement of analyses 'under climate changes conditions' is to strong, in my opinion. Again, there are no analyses of impacts of climate forcing on the physical processes but only a retrospective assessment.

  Authors: The title is changed accordingly: Long-term monitoring (1953-2019) of geomorphologically active sections of Little Ice Age lateral moraines in the context of changing meteorological conditions.

- Reviewer: Also, the literature about the changes in precipitation is wide. Please, have a look to Libertino et al., 2019, Caporali et al.,2021. Within this context, how trends are assessed? Why the choice of those precipitation classes? An example of classification is given by Alpert et al., 2002.

  Authors: As the topographic data, i.e. the available DEMs, resulted in epochs of different lengths, the mean number of events per epoch was calculated and compared accordingly. The choice of these precipitation classes allows a comparison of higher and lower intensities. In contrast to Alpert et al. (2002), we chose an interval of equal size depending on the size of the precipitation classes. This was possible because, in contrast to Alpert et al. (2002), there are generally clearly lower precipitation intensities.

  Reviewer: Uncertainty assessment: the section describes mainly data statistics.

Authors: The chapter has been renamed to "Data statisics".

- Reviewer: Clearly, all the conclusion derived for the meteorological drivers are affected by the modeling uncertainties of the used data. I suggest to smooth the title.

  Authors: The title is changed as already described above: Long-term monitoring (1953-2019) of geomorphologically active sections of Little Ice Age lateral moraines in the context of changing meteorological conditions.

Reviewer: Observations and technical suggestions to improve the manuscript:

- Please, at the end of the introduction, state very clearly which are the scope and the research questions of the work, avoiding to refer to previous works (L126-127).
- Authors: The suggestion for improvement was accepted and changed accordingly. The following section was changed and added: In this study we apply the sediment-contribution-area approach to several LIA lateral moraine sections over several decades and several time periods in the European Central Eastern Alps in order to better understand the paraglacial adjustment process of lateral moraines. Thus, the aim is to find out how the spatial erosion pattern within the areas changes over time. Secondly, we show volume calculations of the entire sites to determine the total sediment yield (and erosion rates). Therefore, by combining high-resolution historical and current DEMs and the corresponding DoDs, we show quantification and analysis of gully system morphodynamics at 12 different sections in the upper reaches of lateral moraines in five different glacier forelands over a total period of several decades (1953-2019) with several survey periods (~1950s to ~1970s, ~1970s to ~2000s and ~2000s to 2017/2019). By using simulated climate data of the glacier forelands we were able to investigate, besides system-internal influences, also external impacts on the morphodynamics, which have not been considered in long-term studies on erosion of LIA lateral moraines so far.
- Reviewer: Since different methods based on data are implemented, I strongly suggest to include a flow chart to clearly describe the entire work flow.
- Authors: After intensive reflection, we would like to omit a flow chart, as we do not see any added value here. Both the topographic data basis and the derivation of the sediment contributing area approach are sufficiently explained.
- Reviewer: Include a list of acronyms (they are impressively a lot!!)
- We have not included a list, but we have chosen to leave the common abbreviations (e.g.: LiDAR, SfM, DEM, DoD) and write out the less familiar ones (e.g..: AoI (to test site/site), sediment yield, catchment area, and sediment contributing area, historical aerial image). Therefore, there are now clearly less abbreviations.
- Reviewer: Sometimes figures and tables captions are not explicative. Please, explain the content of the figure/table in a concise manner.

- Authors: The following table headings and figure captions have been changed:
- Tab. 1: Before: Characteristics of the sites. Values were derived from 2017 DEM (Kaunertal) and 2019 DEM (Horlachtal and Martelltal). After: Location and characteristics of the test sites (Values were derived from 2017 DEM (Kaunertal) and 2019 DEM (Horlachtal and Martelltal)).
- Tab. 3: Before: Overview of the ALS (and DEM) data. After: ALS and DEM data and corresponding flight mission attributes.
- Also Fig. 2, 4, 5 (see below).
- Reviewer: Figure 1: include letters (or number) to refer to the specific subplot figures. At first sight the figure results confusing (AOIs not clearly localized).
- Authors: The request for change has been implemented. The sites are now clearly identifiable.
- Reviewer: Figure 2, 4, 5: detail captions
- Authors: A more detailed caption has been added:

  Fig. 2: Before: Data and time periods. After: Type of topographic dataset and the resulting time periods.

  Fig. 4: Before: Uncertainty assessment. After: Statistical analysis of the uncertainty assessment: (a) Size of the test sites, (b) size of the stabe areas next to the test sites, (c) max. values (stable areas), (d) min. values (stable areas), (e) standard deviation (stable areas), (f) RMSE (stable areas) and (g) arithmetic mean.

  Fig. 5: Before: Example derivation of the statistical relationship at site KG2. After: Derivation of the sediment contributing area approach using the example of test site KG2: (a) Determination of the size of the catchment area, (b) determination of sediment yield, (c) modelling of the channels and exemplary placement of the virtual sediment traps and (d) calculation of the regression lines.

- Reviewer: Figure 6: try to synthesize the results by plotting the slope and intercepts for all the AOIs.
- Authors: These are already in Appendix A.
- Reviewer: Regression is a very simple tool that correlates dependent and independent variables. Please, state clear the problem in L252 to L256! Also lines from L257 to L262 are a bit confusing, I suggest to include a figure (a scheme) to describe the analyzed variables.
- Authors: Figure 5 already explains the variables, so Figure 5 was additionally mentioned in the text. L254 to L267 was removed as this is the original description of the SCA model. Finally, only the SCA approach according to Neugirg et al. (2015; 2016) is shown in the manuscript, which was used in the present study. The problem is now presented right at the beginning.
- Reviewer: L348: provide a reference. Often, other definitions are used.
- Authors: Since the precipitation data were available in a one-hour resolution, they were separated accordingly in the case of no recorded precipitation in the minimum resolution.
- Reviewer: L102: you may have a look to the contribution Noto et al., 2017 for the literature review.
- Authors: The reference suggestion has been read and has been added.
- Reviewer: L124: define AOI

- Authors: Through the review process of the first reviewer, AOI was changed to test site/site, so a definition of AOI is no longer necessary since it no longer appears in the manuscript.
- Reviewer: L194: define HAI
- Authors: In the course of the revision by the first reviewer, HAI (historical aerial image) was written out, so that a definition of HAI is no longer necessary.
- Reviewer: I found some content of the section 5.3.1 (including figure 15) more opportune for the introduction; this concept was not clearly introduced.
- Authors: As the concept introduced is based on the results of this study, it is not yet explained in the introduction, but can be found in the discussion chapter.

- Alpert, P., Ben-Gai, T., Baharad, A., Benjamini, Y., Yekutieli, D., Colacino, M., Diodato, L., Ramis, C., Homar, V., Romero, R., Michaelides, S., and Manes, A.: The paradoxical increase of Mediterranean extreme daily rainfall in spite of decrease in total values, Geophys. Res. Lett., 29, 1–31, doi:10.1029/2001GL013554, 2002.
- Altmann, M., Piermattei, L., Haas, F., Heckmann, T., Fleischer, F., Rom, J., Betz-Nutz, S., Knoflach, B., Müller, S., Ramskogler, K., Pfeiffer, M., Hofmeister, F., Ressl, C., and Becht, M.: Long-Term Changes of morphodynamics on Little Ice Age Lateral Moraines and the Resulting Sediment Transfer into Mountain Streams in the Upper Kauner Valley, Austria, Water, 12, 3375, https://doi.org/10.3390/w12123375, 2020.
- Betz-Nutz, S., Heckmann, T., Haas, F., and Becht, M. 2023. Development of the morphodynamics on LIA lateral moraines in ten glacier forefields of the Eastern Alps since the 1950s, Earth Surf. Dynam. https://doi.org/10.5194/esurf -11-203-2023
- Caporali E, Lompi M, Pacetti T, Chiarello V, and Fatichi S (2021) A review of studies on observed precipitation trends in Italy. Int. J. Climatol., 41, E1–25, TS7. https://doi.org/10.1002/joc.6741.
- Libertino A, Ganora D, Claps P (2019). Evidence for increasing rainfall extremes remains elusive at large spatial scales: The case of Italy. Geophysical Research Letters, 46: 7437– 7446. https://doi.org/10.1029/2019GL083371.
- Noto, L.V., Bastola, S., Dialynas, Y.G., Arnone, E., Bras, R.L., 2017. Integration of fuzzy logic and image analysis for the detection of gullies in the Calhoun Critical Zone Observatory using airborne LiDAR data. ISPRS J. Photogrammetry Remote Sens. 126, 209–2.

---

## Referee Report (RR1)

**Manuscript EGUsphere-2022-1512 - revised version**

**Long-term monitoring (1953-2019) of geomorphologically active sections of Little Ice Age lateral moraines in the context of changing meteorological conditions**

**by Altmann et al.**

**General comments:**

The current manuscript I was invited to review constitutes the revised version of a preprint submitted to ESurf I reviewed myself some time ago. As usual in such cases, I revisited my initial review and primarily focused how the authors addressed it, either by making changes in the manuscript in response to my comments and recommendations or by explaining and elaborating why they disagree with my suggestions and did not or only partly follow them. Additionally, I also had a brief check on the initial comments of the second reviewer to get an overall impression on the quality of the revision performed by the authors.

At first, I am very satisfied that the authors followed my suggestion and in their improved version of the manuscript avoid the excessive use of acronyms and abbreviations initially criticised. The manuscript now reads much more fluently and is much more easy to follow, in particular for readers not familiar with the topic. The authors also followed almost all those other minor technical issues I commented on in my review. Language and structure of the manuscript are now on the level required for final acceptance.

Alongside these technical issues I initially recommend that the authors should extent the discussion chapter by exploring some of their most interesting findings, for example that their investigated active gully systems still are active and show, despite a decrease of sediment yield in most cases, no stabilisation. Another points I suggested to discuss in more detail were the 'sediment activity concept' and a potential future impact of an increase in frequency or magnitude of heavy-precipitation events among the 'meteorological drivers'.

Although the authors did not entirely agree with my suggestions and only partly made some additions in response to them, they explained their reasoning in a convincing fashion and I fully accept their intentions in the context of the submitted manuscript. Because my recommendation to extent the discussion should not have understood as necessary amendment or 'flaw' impacting the scientific quality of the study, also under this aspect the manuscript is now in a form acceptable for publication.

My recommendation is now to accept the revised version for publication.